

# The effects of high versus low talker variability and individual aptitude on phonetic training of Mandarin lexical tones

Hanyu Dong[1], Meghan Clayards[2,3], Helen Brown[4] and Elizabeth Wonnacott[1]

[1] Division of Psychology and Language Sciences, University College London, London, UK
[2] Department of Linguistics, McGill University, Montreal, QC, Canada
[3] School of Communication Sciences and Disorders, McGill University, Montreal, QC, Canada
[4] Department of Psychology, Nottingham Trent University, Nottingham, UK

Corresponding authors
Hanyu Dong,
hanyu.dong.10@ucl.ac.uk
Elizabeth Wonnacott,
e.wonnacott@ucl.ac.uk

## ABSTRACT

High variability (HV) training has been found to be more effective than low variability (LV) training when learning various non-native phonetic contrasts. However, little research has considered whether this applies to the learning of tone contrasts. The only two relevant studies suggested that the effect of HV training depends on the perceptual aptitude of participants (*Perrachione et al., 2011*; *Sadakata & McQueen, 2014*). The present study extends these findings by examining the interaction between individual aptitude and input variability using natural, meaningful second language input (both previous studies used pseudowords). A total of 60 English speakers took part in an eight session phonetic training paradigm. They were assigned to high/low/high-blocked variability training groups and learned real Mandarin tones and words. Individual aptitude was measured following previous work. Learning was measured using one discrimination task, one identification task and two production tasks. All tasks assessed generalization. All groups improved in both the production and perception of tones which transferred to untrained voices and items, demonstrating the effectiveness of training despite the increased complexity compared with previous research. Although the LV group exhibited an advantage with the training stimuli, there was no evidence for a benefit of high-variability in any of the tests of generalisation. Moreover, although aptitude significantly predicted performance in discrimination, identification and training tasks, no interaction between individual aptitude and variability was revealed. Additional Bayes Factor analyses indicated substantial evidence for the null for the hypotheses of a benefit of high-variability in generalisation, however the evidence regarding the interaction was ambiguous. We discuss these results in light of previous findings.

# INTRODUCTION

One challenging aspect of learning a second language (L2) is learning to accurately perceive non-native phonetic categories. This task is particularly difficult when the
L2 relies on the same acoustic dimensions as the first language (L1), but for different purposes (*Bygate, Swain & Skehan, 2013*), suggesting that it is challenging to adjust existing acoustic properties in the L1 to learn new L2 categories. This challenge is compounded by the fact that speech is highly variable in the natural linguistic environment. Variability comes not only from the phonetic context but also from differences between speakers. Thus, learners must learn to distinguish the new L2 categories despite all the variability present in the learning input. There is evidence that native listeners can process this variability in speech faster and more accurately than non-native listeners (*Bradlow & Pisoni, 1999*), indicating that variability is indeed a challenge for L2 learners. Despite this, it has been suggested that input variability may be beneficial for L2 learning and generalisation (*Barcroft & Sommers, 2005*; *Lively, Logan & Pisoni, 1993*). However, recent evidence suggests that the ability to benefit from variability may depend on individual learner aptitude (*Perrachione et al., 2011*; *Sadakata & McQueen, 2014*), at least in the learning of lexical tones (i.e. the distinctive pitch patterns carried by the syllable of a word which, in certain languages, distinguish meaningful lexical contrasts). The current paper further explores how and when variability supports or impedes learning of new L2 phonetic categories, focusing on English learners of Mandarin tone contrasts.

## High variability L2 phonetic training for non-tonal contrasts

A substantial body of literature has explored whether phonetic training can be used to improve identification and discrimination of non-native phonetic contrasts in L2 learners. An early study by *Strange & Dittmann (1984)* attempted to train Japanese speakers on the English /r/-/l/ distinction, a phoneme contrast that does not exist in Japanese. Participants were trained on stimuli from a synthetic *rock-lock* continuum. The key result was that although performance increased both for trained and novel synthetic items, participants failed to show any improvement for naturally produced minimal pair items. Later research suggested that a key factor which prevented generalisation to natural speech tokens was a lack of variability in the training materials: Variability was present in the form of the ambiguous intermediate stimuli along the continuum, however, there was a single phonetic context and a single (synthesized) speaker. *Logan, Lively & Pisoni (1991)* also trained Japanese learners on the English /r/-/l/ contrast, but included multiple natural exemplars spoken by six speakers, with the target speech sounds appearing in a range of phonetic contexts. In contrast to Strange and Dittman, they found that participants successfully generalised both to new speakers and new words at test. This was the first study to indicate the importance of variability within the training materials. A follow up study by *Lively, Logan & Pisoni (1993)* provided further evidence for this by contrasting a condition with *high variability (HV)* input to one with *low variability (LV)* input in which the stimuli were spoken by a single speaker (although still exemplified in multiple phonetic contexts). Participants in the LV group improved during the training sessions but failed to generalise this learning to a new speaker.

Following *Lively, Logan & Pisoni (1993)* *high variability phonetic training* (HVPT) has become standard in L2 phonetic training. This methodology has been successfully extended to training a variety of contrasts in various languages such as learning of

the English /u/-/ʊ/ distinction by Catalan/Spanish bilinguals (*Aliaga-García & Mora, 2009*), learning of the English /i/-/ɪ/ contrast by native Greek speakers (*Giannakopoulou, Uther & Ylinen, 2013*; *Lengeris & Hazan, 2010*), and learning of the English /w/-/v/ distinction by native German speakers (*Iverson et al., 2008*).

There is also some evidence that this type of perceptual training benefits production in addition to perception. *Bradlow et al. (1999)* found that production of the /r/-/l/ contrast improved in Japanese speakers following HVPT, with this improvement being retained even after 3 months. Similar improvement on the production of American English mid to low vowels by Japanese speakers following HVPT was also reported by *Lambacher et al. (2005)*. However, the evidence here is mixed: A recent study by *Alshangiti & Evans (2014)* employed HVPT to train Arabic learners on non-native English vowel contrasts and found no improvements in production, although participants receiving additional explicit production training did show some limited improvement.

Although the studies reviewed above all used HVPT, only the original work by *Lively, Logan & Pisoni (1993)* directly contrasted the use of high and low variability materials. It is notable these seminal experiments used small samples (the tests of generalisation were administered to only three of the participants). Since then, few studies have explicitly contrasted high and low variability training. One such study was *Sadakata & McQueen (2013)*, who trained native Dutch speakers with geminate and singleton variants of the Japanese fricative /s/. Participants were trained with either a limited set of words recorded by a single speaker (LV) or with a more variable set of words recorded by multiple speakers (HV). Both types of training led to increases in generalisation to untrained fricatives and speakers. However, in an identification task, the improvement was greater for participants receiving HV training than those receiving LV training. Similar results were reported by *Wong (2014)* who trained native Cantonese speakers with the English /e/-/æ/ contrast. Both LV (one speaker) and HV (six speakers) training lead to increased performance from pre- to post-test, but the improvement was greater for the HV group. This was found in tests of generalisation to new speakers and new items, and from perception to production. In contrast, a recent phonetic training study did not find the same benefit. *Giannakopoulou et al. (2017)* compared matched HV (four speakers) and LV (one speaker) training for adult and child (8-year-old) native Greek speakers who were trained on the English /i/-/ɪ/ contrast. This study did *not* show a benefit for HV compared to LV training in either age group, even for generalisation items. However, for adult participants, it is unclear the extent to which this was due to ceiling effects. To our knowledge, the only other previous studies that specifically manipulated variability during learning of non-native phonetic categories are those by *Perrachione et al. (2011)* and *Sadakata & McQueen (2014)*, which both looked at the learning of lexical tone. We discuss these studies in more detail in the following section.

Although there is a relatively small evidence base regarding a benefit of high over low phonetic training for non-native phoneme categories, there is further evidence for this benefit in related areas of speech and language learning, specifically accent categorization and adaptation (*Bradlow & Bent, 2008*; *Clopper & Pisoni, 2004*), and L2 vocabulary learning (*Barcroft & Sommers, 2005*, *2014*; *Sommers & Barcroft, 2007*, *2011*). Benefits of

HVPT are generally seen in tasks of generalisation, suggesting that exposure to variation across speakers and/or items boosts the ability to generalise across these dimensions. This intuitively sensible result is in line with the predictions of computational models in which irrelevant contextual/speaker identity cues compete with phonetically relevant cues, so that dissociation of these irrelevant cues is the key mechanism which underpins generalisation (*Apfelbaum & McMurray, 2011*; *Ramscar & Baayen, 2013*; *Ramscar et al., 2010*).

## Phonetic training of L2 lexical tones

Each of the phonetic training studies discussed above involved training a *segmental* contrast (consonantal or vocalic). Lexical tone is another type of phonological contrast in some natural languages, whereby the pitch contour is used to distinguish lexical or grammatical meanings (*Yip, 2002*). For example, Mandarin Chinese has four lexical tones: level-tone (Tone 1), rising-tone (Tone 2), dipping-tone (Tone 3) and falling-tone (Tone 4). These pitch contours combine with syllables to distinguish meanings. For instance, the syllable *ba* combines with the four tones to mean: *eight* (*bā*, Tone 1), *pluck* (*bá*, Tone 2), *grasp* (*b* , Tone 3) and *father* (*bà*, Tone 4). Each of these words thus forms a minimal pair with each of the others. Note that while non-tonal languages such as English use pitch information extensively for intonation (e.g. forming a question, or for emphasis), and that pitch plays a role in marking stress at the lexical level in (e.g. IMport/imPORT), this is quite different from a lexical tone system, causing difficulties for L2 learners of Mandarin.

The first study examining lexical tone training was conducted by *Wang et al. (1999)*. A similar paradigm to that used by *Logan, Lively & Pisoni (1991)* was adopted using four speakers for training. Training materials were all real monosyllabic Mandarin words that varied in the consonants, vowels and syllable structure. During training participants heard a syllable whilst viewing two of the four standard diacritic representations (i.e. →, ↗, ∨, ↘, which are iconic in nature). They were asked to pick out the picture of the arrow that corresponded to the tone. At test, participants chose which tone they had heard out of a choice of all four diacritics. There were also two generalisation tasks, one testing generalisation to untrained items and one testing generalisation to a new speaker. Native American English speakers showed significant improvement in the accuracy of tone identification after eight sessions of HV training over 2 weeks, and this generalised to both new words and a new speaker. In a follow up study, *Wang, Jongman & Sereno (2003)* used the same training paradigm to test whether learning transferred to production. They recruited participants taking Mandarin courses and asked them to read through a list of 80 Mandarin words written in Pinyin (an alphabetic transcription) before and after training. They found improvements in production, although these were mainly seen in pitch contour rather than pitch height.

These studies suggested that as with segmental phoneme contrasts, HV training may also facilitate the learning of tone contrasts. However, *Wang et al. (1999)* and *Wang, Jongman & Sereno (2003)* did not directly contrast high and low variability training materials. *Perrachione et al. (2011)* investigated this contrast directly. They trained native American English speakers with no previous knowledge of Mandarin (or any other tonal

language), using English monosyllabic pseudowords combined with Mandarin tones 1, 2, and 4 (→, ↗ & ↘). The training task used either LV (one speaker) or HV (four speaker) input. During the training, participants matched the sound they heard with one of three pictures of concrete objects presented, where the three words associated with these pictures were minimal trios that differed only in tone. Participants were tested on their ability to generalise their learning to new speakers. Importantly, *Perrachione et al. (2011)* were also interested in the role of individual differences in learning. Therefore, they also determined participants' baseline ability to perceive the tone contrasts prior to training using a *Pitch Contour Perception Test*. In this task, participants heard a vowel produced with either Mandarin tone 1, 2 or 4 whilst viewing pictures of standard diacritics associated with these tones (→, ↗ & ↘), and were asked to select the arrow that corresponded to the tone. Based on performance in this task, the researchers grouped participants into high and low aptitude groups. The results showed that whilst the LV group outperformed the HV group during training (presumably due to accommodation to a repeated speaker throughout the task), there were no differences between the high and LV groups during test. Critically however, there was an interaction between an individuals' aptitude categorization and the type of variability training: Only participants with high aptitude benefitted from HV training, while those with low aptitude actually benefitted more from LV training. It is important to note that this interaction was seen in a task which relied on participants' ability to generalise their learning[1] of tones to an untrained speaker. That is, in a task where we would expect that exposure to multiple speakers would be beneficial since it should allow learners to better dissociate the tones from the particular speakers used in training. These results, therefore, suggest that only the high aptitude learners can take advantage of this benefit. Another training study by *Sadakata & McQueen (2014)* also explored the relationship between input variability and individual aptitude in lexical tone training, though using different training and testing materials. They trained native Dutch speakers (with no prior knowledge of Mandarin or any other tonal language) using naturally produced bisyllabic Mandarin pseudowords. The two syllables in each word either had Tone 2 followed by Tone 1, or Tone 3 followed by Tone 1, and each tone pair was randomly assigned one of two numeric labels (e.g. for one participant Tone 2-Tone 1 was labelled '1', Tone 3-Tone 1 was labelled '2'). During the training task, participants identified the tone pair type of each stimulus by choosing the correct numeric label (e.g. hear /pasa/ with Tone 2-Tone 1, correct response is 1). Thus, in contrast to the study by *Perrachione et al. (2011)*, participants did not need to learn the meaning of each word. Input variability was manipulated, with three levels (low/medium/high). In contrast to the work by Perrachione et al. where the HV and LV conditions differed only in terms of the number of speakers, in this study variability was increased both by including more speakers and more items. Specifically, the number of different vowels used in the bi-syllabic sequences was manipulated: the LV group encountered only one vowel (e.g. pasa, casa, lasa, etc.) whereas the medium and HV groups encountered four different vowels (pasa, pesa, pisa, pusa; casa, cesa, cisa, cusa; lasa, lesa, lisa, lusa etc.). Participants were tested on the trained items (i.e. using trained speakers and trained items). Generalisation was also examined in a number of ways by looking

[1] In their paper, *Perrachione et al. (2011)* do *not* refer to this task as a generalisation task. Instead they report a generalisation measure which is a ratio of performance on this test with novel speakers to performance in training (test-performance/training-performance). Note that this ratio will increase not only if participants are better at test, but also if they are *worse* in training. Using this measure, Perrachione et al. found a benefit of high variability training. However on inspection of the means, it seems that this relationship is driven by the *poorer* performance in training in the high variability condition, rather than by *better* performance in the test with novel speakers. We therefore do not see the ratio measure as providing evidence for an overall benefit of HV training on generalisation.

at (1) trained items spoken by an untrained talker; (2) pseudowords containing untrained vowels (3) pseudowords in which the order of tones in the bi-syllables were reversed (i.e. a novel position), and (4) items where the tone was embedded in a sentence context. As in the study by *Perrachione et al. (2011)*, *Sadakata & McQueen (2014)* also tested individual aptitude but with a different method. They employed a categorization task using stimuli from a six step Tone 2 to Tone 3 continuum (created using natural productions of the two tones with the Mandarin vowel /a/ as endpoints and linearly interpolating between these endpoints). Participants were asked to identify if the sound they heard was more like Tone 2 or Tone 3, and a categorization slope was obtained for each participant providing a measure of their ability to discriminate this contrast, which is generally found to be the most challenging tone contrast for L2 learners of Mandarin. Participants were grouped according to their slopes, and this grouping was entered as a factor in the analyses of tests of learning, along with the effect of training condition (high-medium-low) and the interaction between factors. For the test with trained speakers and items, there was no group level effect of variability condition, however there *was* an interaction between variability and aptitude similar to that reported by Perrachione et al.: Participants with high aptitude benefitted from HV training, while those with lower aptitude benefitted more from LV training. For the generalisation tests, participants showed above chance performance in all but the new position condition, demonstrating an ability to generalise their learning of tone across different dimensions. However, they did *not* demonstrate an overall benefit of higher variability in any of the transfer tests, nor, did variability interaction with aptitude. Note that the overall lack of a HV benefit is again surprising, particularly for test items with untrained talkers and novel items, since the manipulations in training should specifically work to increase generalisation along these dimensions.

In sum, the two studies which have directly compared high and low variability input in training Mandarin tone contrasts have *not* found the predicted benefit of HV on generalisation, either when varying just speakers or when varying speakers and items. However, both of these studies found an interaction between participant aptitude and variability condition. The results of these studies thus provide mutually corroborating evidence—using somewhat different training and testing methods—that the ability to learn from HV input is dependent on learner aptitude, although it should be noted that this interaction was found in a task with untrained speakers in one study (*Perrachione et al., 2011*), but in a task with trained stimuli in the other (*Sadakata & McQueen, 2014*).

Why might the ability to benefit from varied training materials depend on participant aptitude? *Perrachione et al. (2011)* suggest that one reason why low aptitude participants may struggle with multi-speaker input is that the speakers were intermixed during training: This requires trial-by-trial adaptation to each speaker, which was not required in the corresponding single speaker LV conditions. This may place a burden on learners (see *Mattys & Wiget, 2011*; *Nusbaum & Morin, 1992*, for evidence that intermixed multi-speaker stimuli are difficult even for L1 processing and that this interacts with constraints on working memory and attention). To test this, *Perrachione et al. (2011)* conducted a second experiment in which items from each speaker were presented in separate

blocks (as is more common in HVPT). This improved performance during the training task compared with unblocked training for low aptitude learners only, confirming the hypothesis that switching between speakers on a trial-by-trial basis during training interferes with learning for low aptitude learners. On the other hand, *Sadakata & McQueen (2014)* employed a training paradigm in which speakers were blocked in the HV condition, yet they still found the interaction with aptitude. However, recall that in their experiment they also manipulated item variability, yet only speakers were blocked by session, not items. Thus, it remains possible that trial-by-trial inconsistency at the level of items could explain some of the greater difficulty of low aptitude learners in their study.

## The current study

The fact that neither of the tone training studies found an overall benefit of high over LV in tone generalisation is surprising in light of the phonetic literature and the predictions of the computational model (*Apfelbaum & McMurray, 2011*; *Ramscar & Baayen, 2013*) mentioned above. Moreover, as the previous authors point out, if it is actually the case that learning from multiple voices is more or less effective for different groups of learners, this has important implications for the design of L2 training tools. For this to be the case, it is important to establish the generalisability of the findings to different contexts and materials, particularly those which are relevant in an L2 learning context. We suggest that what L2 learners are most interested in developing is their ability to use tone when mapping a word's phonological form to its meaning (and vice versa). In this light, the paradigm used by *Sadakata & McQueen (2014)* lacks ecological validity in looking only at mapping to abstract tone categories. On the other hand, *Perrachione et al. (2011)* do train form-meaning mappings, yet, unlike *Sadakata & McQueen (2014)* they use English pseudo-word stimuli, which has the consequence that learners do not simultaneously have to deal with non-native segments and tones, as in a real world L2 learning situation. Furthermore, although there is limited data on the differences between words and non-words in production, it has been noted that non-words may have different properties from real words even within the same language (*Scarborough, 2012*) and may be more clearly articulated (*Hay, Drager & Thomas, 2013*; *Maxwell et al., 2015*). Thus, using non-words might make stimuli slightly easier to learn than if real words were used.

The current training study addresses these issues in a partial replication of the previous work: We use stimuli produced by native Mandarin speakers which are real words in that language. This design choice follows earlier studies such as *Wang et al. (1999)* using a paradigm in which participants are trained to identify word meaning on the basis of tone. In contrast to the previous studies, we also trained the contrasts between all four tones (six tone contrasts) rather than just three (on the assumption that learners are interested in learning the complete set of contrasts within a particular language). We note that these design choices potentially increase the difficulty of our training materials compared to previous work. A key question was whether these choices would impact the interaction between learner aptitude and the benefits of more variable training materials.

We followed *Perrachione et al. (2011)* in varying variability along one dimension only—speaker variability, keeping training items identical across conditions. We also
followed *Perrachione et al. (2011)* in comparing HV input which was blocked by speaker, with input that was not, making three training conditions: LV (one speaker), HV (four speakers intermixed within each training session) and blocked training (four speakers each presented in separate blocks). Note that our choice to manipulate only talker-variability means that the HV blocked condition is matched to the LV condition in terms of trial-by-trial inconsistency, unlike in *Sadakata & McQueen (2014)* where, even though they blocked by speaker, the HV condition contained more trial-by-trial variability in terms of items. We predicted that the difficulty of HV input for lower aptitude participants would be greater in the unblocked condition, thus potentially increasing the likelihood of seeing the predicted interaction between variability and learner aptitude. On the other hand, blocked input is more usual of HVPT (*Iverson, Hazan & Bannister, 2005*; *Logan, Lively & Pisoni, 1991*) and may increase the possibility of seeing an overall benefit of speaker variability on generalisation.

We used two perceptual tasks designed to tap individual aptitude. These were adapted from those used in *Perrachione et al. (2011)* and *Sadakata & McQueen (2014)*. However, while the previous studies grouped participants into one of two categories (high aptitude vs low aptitude) based on the aptitude score, in the current study they were used as continuous measures. This allowed us to avoid assigning an arbitrary 'cut off' for high vs low aptitude groups, and the loss of information which occurs when an underlying continuous variable is turned into a binary measure. Note that the statistical approach used in the current paper (logistic mixed effect models) allowed us to include continuous predictors and look at their interactions with other factors.

A further extension in the current study is that we use several new outcome measures to test learning and generalisation. First, most similar to the task used in *Perrachione et al. (2011)* was a picture identification task which was a version of the training task (2AFC picture identification) without feedback. Following *Perrachione et al. (2011)* we included untrained-speaker items, where benefits of speaker variability in training should be most apparent. However, bearing in mind that *Sadakata & McQueen (2014)* actually found the key interaction with aptitude *only* in the test with trained stimuli, we also included trained-speaker test items.

We also included a second perceptual task which did *not* involve knowing specific form-meaning mappings and thus had the benefit that it could be conducted both pre- and post-test. This was a three interval oddity task which required participants to pick the odd-one-out after hearing three words spoken aloud, each by a different speaker. Two of the tokens were productions of the same word and the third differed only in the tone (e.g. *bā*, Tone 1; *bā*, Tone 1; *bà*, Tone 4). Because all three tokens are physically different, it requires the listener to focus on the phonological level ignoring irrelevant acoustic differences. Furthermore, the use of three speakers forces the listener to ignore irrelevant speaker-specific differences, making it especially challenging (*Strange & Shafer, 2008*). This task used untrained speakers in every trial, so that every test-item required generalisation to new speakers[2]. In addition, here it was possible to use both trained and untrained *items*. Note that even though the variability over items is matched across conditions, it is possible that varying speaker specific cues might also thus promote

[2] If we wished to use trained speakers, in order to be able to the use the same test with the low variability condition, we would have to use a single speaker across all three test trials. Our pilot work suggested that participants performed at ceiling on a single-speaker version of this task, even at pre-test.

**Table 1 Mean age range, average number of languages learned and mean starting age of learning the first L2 for participants in each condition.**

| Condition | Mean age | Age range | Languages learned | Average staring age |
|---|---|---|---|---|
| Low variability | 26.15 (2.2) | 19–53 | 2.7 (0.5) | 13.8 (1.1) |
| High variability | 25.65 (0.7) | 19–47 | 2.5 (0.6) | 12.2 (0.5) |
| High variability blocked | 22.05 (1.4) | 19–30 | 2.0 (1.3) | 11.8 (0.4) |

generalisation across this dimension. If this is the case, a HV benefit may be stronger for untrained items than trained items.

Finally, we also tested production using a picture naming task at post-test, in which participants were required to name the pictures used in training in Mandarin. We also conducted a word repetition task, which had the benefit that it could also be conducted at pre-test, and that we could use both trained and untrained words (as for the three-interval oddity task discussed above). Although there is evidence HVPT can benefit the production of tones (*Wang, Jongman & Sereno, 2003*), there has been no direct examination of whether HV training materials are more effective than LV training materials for production. However, more generally in the L2 vocabulary learning literature, training with multiple speakers has been found to lead to better recall in a picture naming task (*Barcroft & Sommers, 2005*), suggesting that the HVPT advantage should extend to production measures.

In sum, the current experiment assessed whether individuals benefit from high over LV perceptual training when learning novel L2 tone contrasts, and whether this interacts with learner aptitude. We used measures of aptitude taken from previous studies, but a training paradigm with real Mandarin stimuli embedded in a vocabulary learning task, which trained discrimination of all six Mandarin tone contrasts. Learning and generalisation were measured in multiple tests of both perception and production. In general, the current design increased ecological validity and likely also increased the difficulty of the learning task relative to previous work. It is possible that increasing difficulty could exacerbate differences between learners of different aptitudes, potentially increasing the effect. On the other hand, it is also possible that the increased difficulty might make HV input much harder for all participants, decreasing or removing the specific benefit of HVPT for high aptitude learners.

## METHOD

### Participants

A total of 60 adults recruited from UCL Psychology Subject Pool participated in the experiment, 20 in each of the three conditions (LV, HV, high variability blocked (HVB)). Participant information is summarized in Table 1. There was no difference between these groups in age, $F(2,57) = 1.95$, $p = 0.15$. Participants had no known hearing, speech, or language impairments. Written consent was obtained from participants prior to the first session. Each participant was paid £45 at the end of the study.

**Table 2 Use of trained and untrained items and voices in different tasks.**

| Task | Items | Voice |
|---|---|---|
| Picture identification | Trained | One trained voice (counterbalanced, see Table 3) One untrained voice (counterbalanced, see Table 3) |
| Three interval oddity (Pre and Post) | Trained and untrained | Four new voices |
| Picture naming | Trained | NA |
| Word repetition (Pre and Post) | Trained and untrained | One trained voice (counterbalanced, see Table 3) |
| Individual aptitude test 1 Pitch contour perception test (Pre and post) | Vowels | Four untrained voices |
| Individual aptitude test 2 Categorisation of synthesised tonal continua (Pre and Post) | Synthesised voice | Synthesised voice |

All participants except three were native English speakers. Of the remaining three, one participant (LV condition) was a native bilingual of English and Hindi, one participant (HV condition) was a native French speaker, and one participant (HV condition) was a native Finnish speaker. Critically, participants had no prior experience of Mandarin Chinese or any other tonal language. On average, participants had learned 2.4 (SD = 0.8) languages and the average age for starting to learn the first L2 was 12.6 years (SD = 1.3).

Ethical approval was given by the UCL Research Ethics Committee with the approval number 6176/002.

## Stimuli

### Stimuli used in training and in the picture identification, three interval oddity, word repetition and picture naming tests

These stimuli consisted of 36 minimal pairs of Mandarin words (six minimal pairs for each of the six tone contrasts generated by the four Mandarin tones). The words in each pair contained the same phonemes, differing only in tone (e.g. *māo*, Tone 1 (*cat*) vs *mào*, Tone 4 (*hat*)). All words were picturable and started with a wide range of phonemes (see Appendix A). In order to examine generalisation across items, half of the word pairs (three per tone contrast) were designated 'trained' words and other half were designated 'untrained' words. Trained words were encountered in both training and test tasks; untrained words were only encountered in the three interval oddity and word recognition tests.

The full set of 72 Mandarin words was recorded by two groups of native Mandarin speakers using a Sony PCM-M10 handheld digital audio recorder. The first group consisted of three female and two male speakers. These stimuli were used in the Training, Word Repetition and Picture Identification tasks. The second group consisted of three new female speakers and two new male speakers. These stimuli were used in the three interval oddity task (making all new speakers in that task). See Table 2 for a summary of the manipulation of item and speaker novelty across the different test tasks, and Table 3 for the tasks in which speakers are counterbalanced.

In the LV condition only one speaker (Trained voice 1) was used in training, and this same speaker was also used as the test voice in the Word Repetition test and for trained

**Table 3 Counterbalancing of voices across training conditions in the picture identification task (the only test in which trained and untrained voices are directly contrasted) and the Word Repetition tests.**

| Task | Voice | | | | |
|------|-----------|-----------|-----------|-----------|-----------|
| | Version 1 | Version 2 | Version 3 | Version 4 | Version 5 |
| Training, LV | F1 | F2 | F3 | M1 | M2 |
| Training, HV/HVB | F1 | F2 | F3 | M1 | M2 |
| | F3 | F1 | M2 | F1 | F2 |
| | M1 | M1 | F1 | F2 | F3 |
| | M2 | M2 | F2 | F3 | M1 |
| Picture Identification | | | | | |
|   Trained voice | F1 | F2 | F3 | M1 | M2 |
|   Untrained voice | F2 | F3 | M1 | M2 | F1 |
| Word repetition | F1 | F2 | F3 | M1 | M2 |

items in the picture identification test. In the HV conditions, four speakers (Trained voice 1 plus three others) were used in training. Only one of these speakers (Trained voice 1) was used in the word repetition test and for trained items in the picture identification test. In all conditions, a further speaker (Untrained voice 1) was assigned to the untrained test items in the picture identification test. The assignment of speakers was rotated across participants, resulting in five counterbalanced versions of each condition (see Table 3). This ensured that any difference found between the low and HV conditions, and between trained and untrained voices, were not due to idiosyncratic difference between speakers. There was no counterbalancing of speaker in other tasks.

All words were edited into separate sound files, and peak amplitude was normalised using Audacity (*Audacity Team, 2015*, http://audacity.sourceforge.net/). Any background noise was also removed. All recordings were perceptually natural and highly distinguishable as judged by native Chinese speakers. Clipart pictures of the 72 words were selected from free online clipart databases.

### Stimuli used in the aptitude tests

Pitch Contour Perception Test: Six Mandarin vowels (/a/, /o/, /e/, /i/, /u/, /y/) were repeated in the four Mandarin tones by two male and two female native Mandarin speakers from talker set 2, making 96 stimuli in total. Stimuli were identical across conditions and participants.

Categorization of Synthesised Tonal Continua: Natural endpoints were chosen from a native Mandarin male speaker producing the word '*wan*' with both Tone 2 and Tone 3. A neutral vowel was also recorded by a native male English speaker producing the 'father vowel' /a/. This vowel was edited slightly to remove portions containing creaky voice at the end. The three syllables (wan (Tone 2), wan (Tone 3), /a/) were then manipulated in Praat (*Boersma & Weenink, 2015*). All three syllables were normalised to be approximately 260 ms long using the Pitch Synchronous Overlap and Add method. The neutral vowel was manipulated to have a flat fundamental frequency (148 Hz) and a flat intensity contour (75 dB). The pitch contours of the two natural endpoints were extracted and a

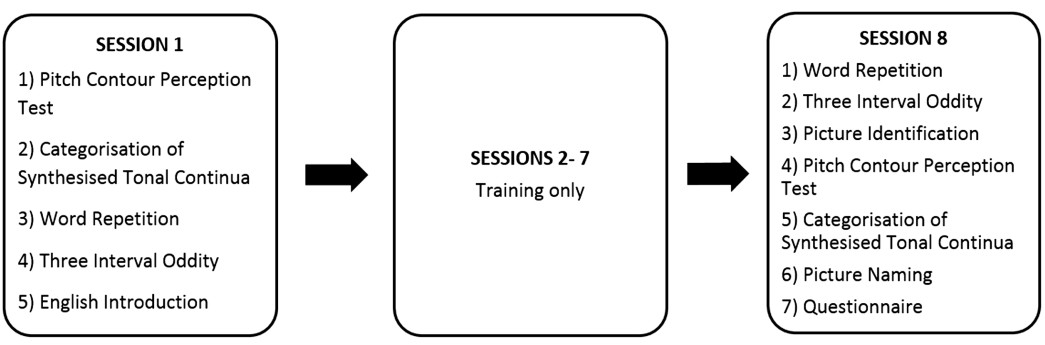

**Figure 1 Tasks completed in each of the eight sessions.** This figure describes all tasks arranged through session 1–8.                         

six-step pitch continuum (Step 1: Tone 2, Step 6: Tone 3) was generated by linearly interpolating between the endpoints. These six pitch contours were then each superimposed on a copy of the neutral vowel using the PSOLA method. Stimuli were identical across participants and conditions.

## Procedure

The experiment involved three stages (see Fig. 1): Pre-test (session 1), training (sessions 2–7), and post-test (session 8). Participants were required to complete all eight sessions within 2 weeks, with the constraint of one session per day at most. The majority of sessions took place in a quiet, soundproof testing room in Chandler House, UCL. The remaining sessions took place in a quiet room in a student house.

Participants were given a brief introduction about the aim of the study and told that they were going to learn some Mandarin tones and words. They were explicitly told that Mandarin has four tones (flat, rising, dipping and falling) and that the tonal differences were used to distinguish meanings. The experiment ran on a Dell Alienware 14R laptop with a 14-inch screen. The experiment software was built using a custom-built software package developed at the University of Rochester.

The specific instructions for each task were displayed on-screen before the task started. After each task, participants had the opportunity to take a 1-min break. The tasks completed in each session are listed in Fig. 1 and described in more detail below. Note that the Pitch Contour Perception Test and Categorisation of Synthesised Tonal Continua were carried out at the beginning of the first session as they provided the measure of individual aptitude prior to exposure to any Mandarin stimuli. There was no time limit for making responses in any of the tasks. Participants wore a pair of HD 201 Sennheiser headphones throughout the experiment with audio stimuli presented at a comfortable listening level.

### Individual aptitude measures

#### The pitch contour perception test
This test was based on the work of *Wong & Perrachione (2007)*. Participants heard a tone (e.g. /a/ (Tone 1)), while viewing pictures of four arrows indicating the different pitch contours. Participants clicked on the arrow that they thought matched the tone heard.

No feedback was provided. There were 96 stimuli in total (four speakers * four tones * four vowels). This task provided another measure of individual differences in tone perception prior to training. Although Perrachione et al. only conducted this task at pre-test, for consistency with the Categorization of Synthesised Tonal Continua (described below) we also repeated the test at post-test and conducted analyses to identify whether performance on this task was itself improved as a result of training (see section 'Categorisation of Synthesised Tonal Continua').

*Categorisation of synthesised tonal continua*
This test was based on Sadakata & McQueen (2014). Participants first practiced listening to Tone 2 and Tone 3 while viewing the corresponding picture of an arrow depicting the pitch change. Each tone was repeated 10 times. In each test trial, participants then decided whether the sound they heard was closer to Tone 2 or Tone 3 by clicking on the corresponding arrow. No feedback was provided. The speech continua consisted of six steps (Step 1: Tone 2, Step 6: Tone 3) with each step repeated 10 times per block. Participants completed two blocks, with an optional 1 min break in the middle, resulting in 120 trials in total. This task provided a measure of individual differences in tone perception prior to training. In line with Sadakata and McQueen's procedure, participants completed the task both before and after training and we conducted analyses to explore whether there was improvement from pre to post-test (section 'The Pitch Contour Perception Test').

**Training task**
Participants completed the training task in Session 2–7. On each trial, participants heard a Mandarin word and selected one of two candidate pictures displayed on the computer screen. The two pictures always belonged to the same minimal pair. Feedback was provided about whether the answer was correct (a green happy face appeared) or incorrect (a red sad face appeared). If the correct choice was made, a picture of a coin also appeared in a box on the left-hand side of the screen, with the aim of motivating participants to try to earn more coins in each subsequent session of training. After that, everything but the correct picture was removed from the screen and the participant heard the correct word again. In the lower right corner of the screen a trial indicator of X/288 was displayed where X indicated the number of trials completed. This tool helped participants to keep track of their performance (see Fig. 2).

There were 18 picture/word pairs used. Each word was used as the target four times. Thus, each picture pair appeared eight times, resulting in 288 trials per session. Participants were assigned to one of the following conditions: LV, HV and HVB (with the assignment of speakers counterbalanced—see Table 3). Each training session lasted for approximately 30 min.

In the LV condition, only *one* speaker was used. In the HV conditions, *four* speakers were used. For each participant, each of their six training sessions was identical. In the HV condition without blocking, all of the speakers were heard in each of the training sessions, with the order randomised so that speaker varied from trial to trial. In contrast,

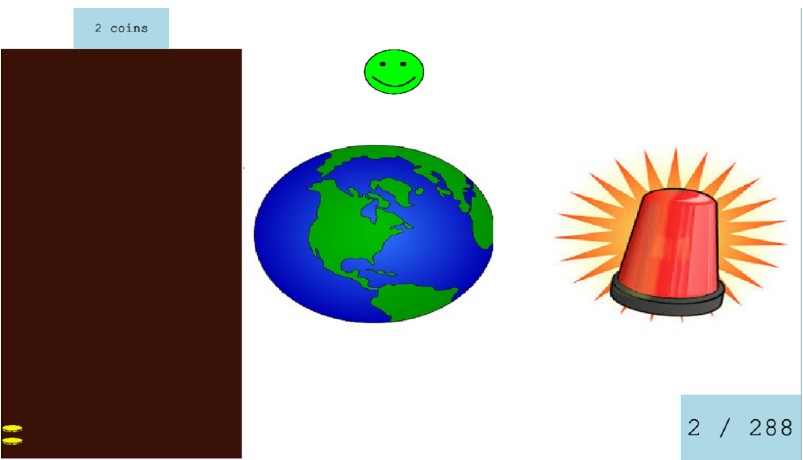

**Figure 2 Screen shot from the Training task.** The stimuli heard is 'dì', tone 4, (earth). The foil picture on the right is 'dí' tone 2, (siren).

in the HV blocked condition, from Day 1 to Day 4 of training (i.e. Session 2–5), only one speaker was involved on each day's training session, (with the trained speaker that was used in the test tasks (e.g. F1 for Version 1) always occurring on Day 3 (i.e. Session 4)); on Days 5 and 6 of training (i.e. Sessions 6 and 7), participants heard all four speakers, each in a separate block, with each word being repeated twice in each voice on these days. In all three conditions, the order of items was randomised within each session.

## Perceptual tests

### Three interval oddity test (pre-post test)

This task required participants to identify the odd one out (i.e. the stimulus with a different tone) from a choice of three Mandarin words, each spoken by a different speaker. Four untrained speakers were used (three female, one male). Each trial used one of the 36 minimal pairs from the main stimuli set (18 trained pairs, 18 untrained pairs). Preliminary work suggested that trials differed in difficulty depending on whether the 'different' stimulus was spoken by the single male speaker, or one of the three female speakers. We therefore ensured that there were equal numbers of the following trial types: (i) 'Neutral'—all three words were spoken by female speakers (ii) 'Easy'—the 'different' word was spoken by a male speaker and the other two were spoken by female speakers; (iii) 'Hard'—the 'different' word was spoken by a female speaker and the other two were spoken by one male speaker and one female speaker. Each of the words in the minimal pair was used once as the target ('different') word, making 72 trials in total.

During the task, three frogs were displayed on the screen. Participants heard three words (played with ISIs of 200 ms) and indicated which word was the odd one out by clicking on the appropriate frog, which could be in any of the three positions. They could not make their response until all three words had been heard, at which point a red box containing the instruction 'Click on the frog that said the different word' appeared at the bottom of the screen. No feedback was provided. Participants completed this task twice—once in the pre-test, and once in the post-test.

*Picture identification test (post-only test)*

This task was the same as the training task with the following changes. Firstly, each word was only repeated twice, once by a trained speaker (trained voice 1) and once by an untrained speaker (Untrained voice 1), making 72 trials in total. Secondly, no feedback was given. This task was completed only in the post-test.

### Production test

*Word repetition test (pre-post test)*

All 72 Mandarin words from the main stimulus set (18 trained pairs, 18 untrained pairs) set were presented one at a time in a randomised order. They were always spoken by the same speaker and this speaker was also used in their training stimuli (training voice 1; see Table 3). After each word, 2 s of white noise was played. This was included to make sure that participants had to encode the stimulus they were repeating and could not access the information in echoic storage (*Flege, Takagi & Mann, 1995*). Participants were instructed to listen carefully to the word and then to repeat the word aloud after the white noise. Verbal responses were digitally recorded and were later transcribed and rated by native speakers of Mandarin (see section 'Coding and Inter-rater Reliability Analyses'). This task was completed once in the pre-test and once in the post-test.

*Picture naming test (post-only test)*

All 36 pictures from the training words were presented in a randomised order. Participants were instructed to try to name the picture using the appropriate Mandarin word. Verbal responses were recorded and were later transcribed and rated by native Mandarin speakers (see section 'Coding and Inter-rater Reliability Analyses'). This task was completed only in the post-test.

### Other tasks

*English introduction task*

This task was included in the batch of tasks administered at pre-test in case the meaning of some pictures were ambiguous (not all items were concrete nouns—for example, '*to paint*'). Participants saw each of the 36 pictures from the training set presented once each in a random order and heard the corresponding English word. No response was recorded. Participants completed this task only once, at the end of the pre-test session.

*Questionnaires*

Participants completed a language background questionnaire after the experiment. Participants were asked to list all the places they had lived for more than 3 months and any languages that they had learned. For each language the participant was asked: (a) to state how long they learned the language for and their starting age; (b) to rate their own current proficiency of the language.

## RESULTS

### Statistical approach

Three different sets of frequentist analyses are reported. First, we conducted the analysis on two individual aptitude measures Categorisation of Synthesised Tonal Continua and Pitch

Contour Perception Test. The primary aim of these analyses was to ensure that the three groups did not differ at pre-test, however we also looked for possible differences at post-test. Second, separate analyses are reported on data from the tests administered pre- and post-training (i.e. Word Repetition task and Three Interval Oddity task), the data collected during Training and the data from the two tasks administered only at post-test (i.e. the Picture Identification task and Picture Naming task). These analyses explored the effects of our experimentally manipulated conditions on the various measures of Mandarin tone learning. Third, analyses were conducted exploring the role of aptitude in each of these tasks (section 'Analyses with Individual Aptitude'). Specifically, we wanted to see whether aptitude interacted with *variability-condition* in predicting the benefits of training, in line with the predictions of previous research (*Perrachione et al., 2011*; *Sadakata & McQueen, 2014*).

Except where stated, analyses used logistic mixed effect models (*Baayen, Davidson & Bates, 2008*; *Jaeger, 2008*; *Quené & Van den Bergh, 2008*) using the package lme4 (*Bates et al., 2013*) for the R computing environment (*R Development Core Team, 2010*). Logistic mixed effect models allow binary data to be analysed with logistic models rather than as proportions, as recommended by *Jaeger (2008)*. In each of the analyses, the factor *variability-condition* has three levels (LV, HV and HVB) which we coded into two contrasts with LV as the baseline (LV vs HV, LV vs HVB). An exception to this is the training data, where a model containing all three conditions would not converge and we took a different approach, as described in the section 'Training'. We also included the interactions between these contrasts and the other factors. We used centred coding which ensured that other effects were evaluated as averaged over all three levels of *variability-condition* (rather than the reference level of LV[3]). Similarly, for the Three Interval Oddity task, we included a *trial-type* factor. The purpose of this was to control for the fact that participants were likely to find some trial types easier than others due to the gender of the speakers producing the stimuli (see section 'Three Interval Oddity Test (Pre-Post Test)'). We therefore coded a factor *trial-type* with three levels (neutral, easy, hard–see method) and included contrasts with neutral ('neutral vs easy' and 'neutral vs hard') using centred coding. In order to perform the analysis comparing pre- and post-test performance, *test-session* was coded as a factor with two levels (pre-test/post-test) with 'pre-test' set as the reference level. This allowed us to look at the (accidental) possible differences between the experimental conditions at the pre-test stage, as well as whether post-test performance differed from this baseline. All other predictors, including both discrete factor codings with two levels (*item-novelty* in the Word Repetition and Three Interval Oddity tasks, and *voice-novelty* in the Picture Identification task) and numeric predictors (*training-session*) in the Training data analyses and the individual difference measures in the models reported in the section 'Analyses with Individual Aptitude'), were centred (i) to reduce the effects of collinearity between main effects and interactions, and (ii) so that the main effects were evaluated as the average effects over all levels of the other predictors (rather than at a specified reference level for each factor). We automatically put experimentally manipulated variables and all of their interactions into the model, without using model selection (except for *trial-type* in the Three Interval

[3] This differs from the default coding of contrasts in the lme4 package. It was achieved by replacing the three-way factor 'condition' with two centred dummy variables and using the main fixed effects from the output of this model.

 

Oddity task which works as a control factor and for this factor we only used its main effect and the interaction with *test-session*). However, we did not inspect the models for all main effects and interactions. Instead, we report the statistics which were necessary to look for accidental differences at pre-test, and those related to our hypotheses. We aimed to examine whether the training improved participants' performance on both untrained items and untrained voices and whether such improvement was modulated by their individual aptitudes. Participant is included as a random effect and a full random slope structure was used (i.e. by-subject slopes for all experimentally manipulated within-subject effects (*test-session*, *voice-novelty*, *item-novelty*) and interactions, as recommended by *Barr et al. (2013)*. In some cases the models did not converge and in those cases correlations between random slopes were removed. Models converged with bound optimization by quadratic approximation (BOBYQA optimization; *Powell, 2009*). R scripts showing full model details can be found here: https://osf.io/wdh8a/.

In addition to the frequentist analyses, in order to aid interpretation of key null results we also included Bayes factor analyses. Our approach for these is described within the relevant section (Section 'Bayes Factor Analyses').

## Individual aptitude tasks

### The pitch contour perception test

The predicted variable was whether a correct response was given (1/0) on each trial. The predictors were the contrasts between *variability-conditions* (LV vs HV; LV vs HVB) and *test-session* (pre-test, post-test). There was no significant difference between the LV and HV groups ($\beta = -0.35$, SE = 0.26, $z = -1.38$, $p = 0.17$) or between the LV and HVB groups ($\beta = 0.17$, SE = 0.26, $z = 0.66$, $p = 0.51$) at pre-test on this measure. Participants showed significant improvement after training ($\beta = 0.21$, SE = 0.05, $z = 4.13$, $p < 0.001$), which can be seen in Fig. 3.

Thus, the three participant groups did not differ in their pre-test performance and the groups showed equivalent improvement from pre- to post-test. Given that this measure is affected by training, we used participants scores at pre-test as our measure of individual differences in the analyses reported in the section 'Analyses with Individual Aptitude'.

### Categorisation of synthesised tonal continua

We estimated individual's performance on the Categorisation of Synthesised Tonal Continua task following *Sadakata & McQueen (2014)*. We used the Logistic Curve Fit function in SPSS to calculate a slope coefficient for each participant (*Joanisse et al., 2000*). The slope (standardised $\beta$) indicates individual differences in tone perception. The smaller the slope, the better the performance. Sadakata and McQueen, removed data from participants with a slope measuring greater than 1.2. Using this threshold 43/60 participants failed the threshold in the current study. This is consistent with the observation that most of the participants were not able to consistently categorise the endpoints of the continua, indicating that this was not a good test of aptitude. We do not report further analyses involving this aptitude variable however they can be found in the supplemental materials (https://osf.io/wdh8a/).

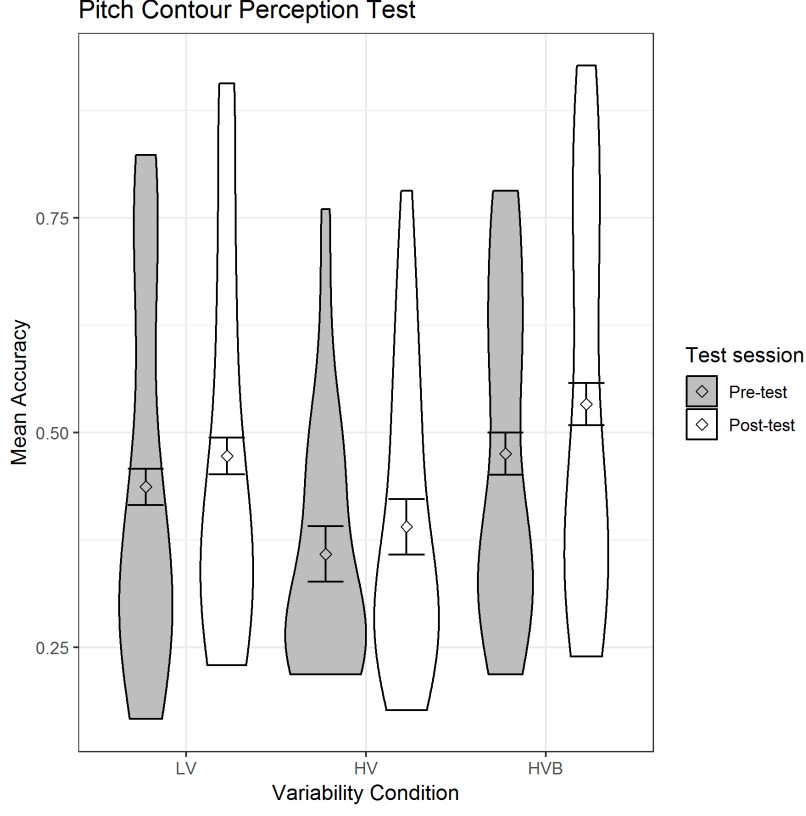

**Figure 3** Mean accuracy for the LV (low variability), HV (high variability) & HVB (high variability blocked) groups in Pitch Contour Perception Task. Error bars represents the 95% confidence intervals.

## Training

A model containing data from all three conditions did not converge; however two separate models, one including the LV and HV conditions, and the other the LV and HVB conditions (with condition as a factor with two levels), did converge. In each case the predicted variable was whether a correct response was given (1/0) on each trial.

The predictors were the numeric factor *training-session* (1:6) and the factor *variability-condition* which had two levels (Model 1: LV vs HV; Model 2, LV vs HVB). The mean accuracy is displayed in Fig. 4.

In both models, there was an effect of *training-session* (Model 1: $\beta = 0.49$, SE = 0.04, $z = 11.52$, $p < 0.001$; Model 2: $\beta = 0.53$, SE = 0.04, $z = 12.17$, $p < 0.001$): Participants' performance increased significantly over time, with additional training sessions. Overall, the LV group performed better than both the HV group ($\beta = -0.79$, SE = 0.16, $z = -5.03$, $p < 0.001$) and the HVB group ($\beta = -0.83$, SE = 0.32, $z = -2.61$, $p < 0.01$). However, the LV vs HV contrast was also modulated by an interaction with *test-session* ($\beta = -0.19$, SE = 0.04, $z = -4.59$, $p < 0.001$), as was the LV vs HVB contrast ($\beta = -0.35$, SE = 0.08 $z = -4.33$, $p < 0.001$). From Fig. 4 it can be seen that the LV and the HVB group did not differ in the first session (i.e. where they get identical input) but the difference gradually

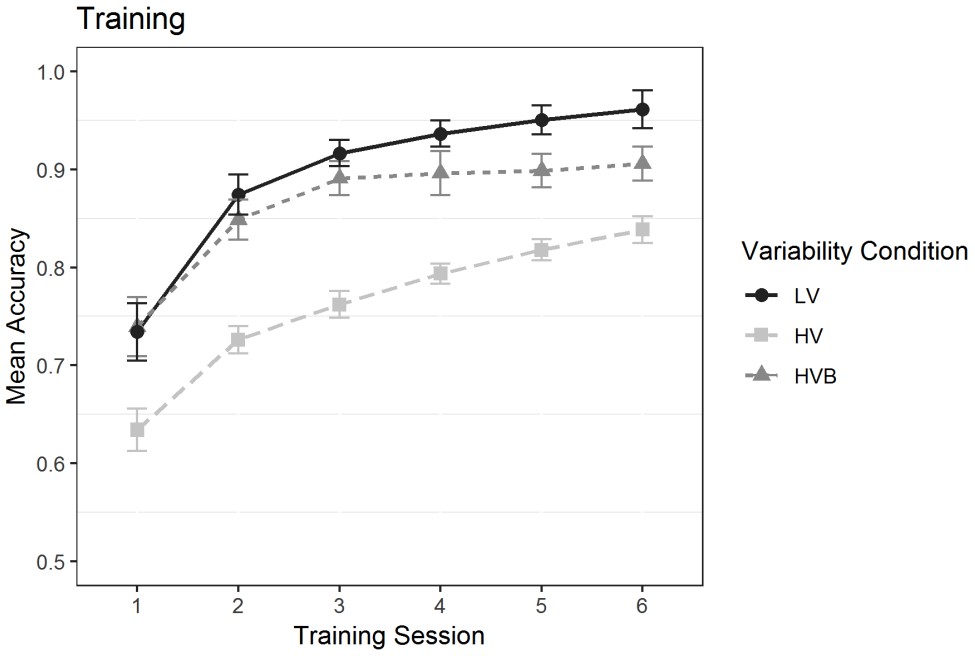

**Figure 4 Mean accuracy in the Training task for the LV (Low Variability), HV (High Variability) and HVB (High Variability Blocked) training groups in each session. Y-axis starts from chance level.** Error bars show 95% confidence intervals.

increased over the next few sessions. For the LV and the HV group, they differed starting from the first session and this difference continued to increase throughout training.

## Perceptual tests

### Three interval oddity task

The predicted variable was whether a correct response was given (1/0) on each trial. The predictors were *test-session* (pre-test, post-test), *variability-condition* (LV vs HV, LV vs HVB), *trial-type* (neutral vs easy, neutral vs hard) and *item-novelty* (trained item, untrained item). The mean accuracy is displayed in Fig. 5.

At pre-test, there was no significant difference between the LV and HV groups ($\beta = -0.002$, SE = 0.14, $z = -0.01$, $p = 0.99$) nor between the LV and HVB groups ($\beta = 0.12$, SE = 0.14, $z = 0.86$, $p = 0.39$), suggesting that the groups started at a similar level. However, performance with the 'untrained' was significantly greater than performance on the 'trained' items at pre-test ($\beta = -0.31$, SE = 0.06, $z = -4.95$, $p < 0.01$), suggesting incidental differences between item sets. As expected, at pre-test participants performed significantly better on 'easy' trials (where the target speaker had a different gender) than 'neutral' trials (where all three speakers had the same gender, $\beta = 0.40$, SE = 0.08, $z = 5.09$, $p < 0.01$) and 'neutral' trials were marginally easier than 'hard' trials (where one of the foil speakers had the odd gender out, $\beta = -0.14$, SE = 0.08, $z = -1.81$, $p = 0.07$).

Overall, participants' performance increased significantly after training ($Mpre = 0.59$, $SDpre = 0.21$, $Mpost = 0.66$, $SDpost = 0.19$, $\beta = 0.31$, SE = 0.05, $z = 6.54$, $p < 0.001$). The interaction between *test-session* and *item-novelty* was not significant ($\beta = 0.14$, SE = 0.09, $z = 1.49$, $p = 0.14$), suggesting no evidence that training had a greater effect

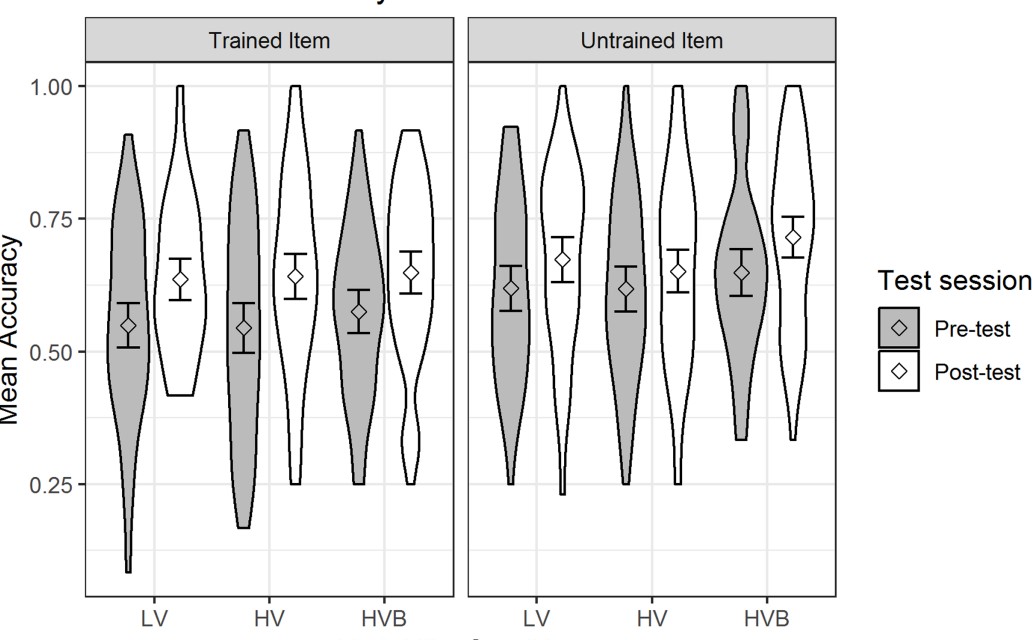

**Figure 5 Mean accuracy in Three Interval Oddity task for LV (low variability), HV (high variability) and HVB (high variability blocked) training groups in Pre- and Post-tests for trained and untrained items.** Error bars show 95% confidence intervals.

for trained words than for untrained words. Critically, there was no interaction with *test-session* for either the contrast between the LV vs the HV conditions ($\beta = -0.01$, $SE = 0.12$, $z = -0.12$, $p = 0.90$) or the contrast between the LV vs the HVB conditions ($\beta = 0.01$, $SE = 0.12$, $z = 0.11$, $p = 0.91$) and they were not qualified by any higher level interactions with *item-novelty* (LV vs HV: $\beta = -0.1$, $SE = 0.22$, $z = -0.64$, $p = 0.52$; LV vs HVB: $\beta = 0.13$, $SE = 0.22$, $z = 0.57$, $p = 0.57$). This suggests no evidence that the extent to which participants improved on this task between pre and post-test differed according to *variability-conditions*, or that this differed for *trained* vs *untrained* items.

Although not part of our key predictions, we also looked to see if there was evidence that participants improved more with the easier or harder trials. In fact, the interaction between *test-session* and the contrast between 'easy' and 'neutral' was significant ($\beta = -0.27$, $SE = 0.11$, $z = -2.39$, $p = 0.02$) while the contrast between 'neutral' and 'hard' was not ($\beta = 0.12$, $SE = 0.11$, $z = 1.06$, $p = 0.29$). This was due to the fact that there was improvement for 'neutral' ($Mpre = 0.57$, $SDpre = 0.14$, $Mpost = 0.65$, $SDpost = 0.15$) and 'hard' trials ($Mpre = 0.54$, $SDpre = 0.16$, $Mpost = 0.65$, $SDpost = 0.15$) but not for 'easy' trials ($Mpre = 0.66$, $SDpre = 0.16$, $Mpost = 0.68$, $SDpost = 0.15$).

### Picture identification

The predicted variable was whether a correct response was given (1/0) on each trial. The predictors were the factor *voice-novelty* (Trained voice, Untrained voice) and the factor *variability-condition* which had two contrasts (LV vs HV, LV vs HVB). The mean accuracy is displayed in Fig. 6.

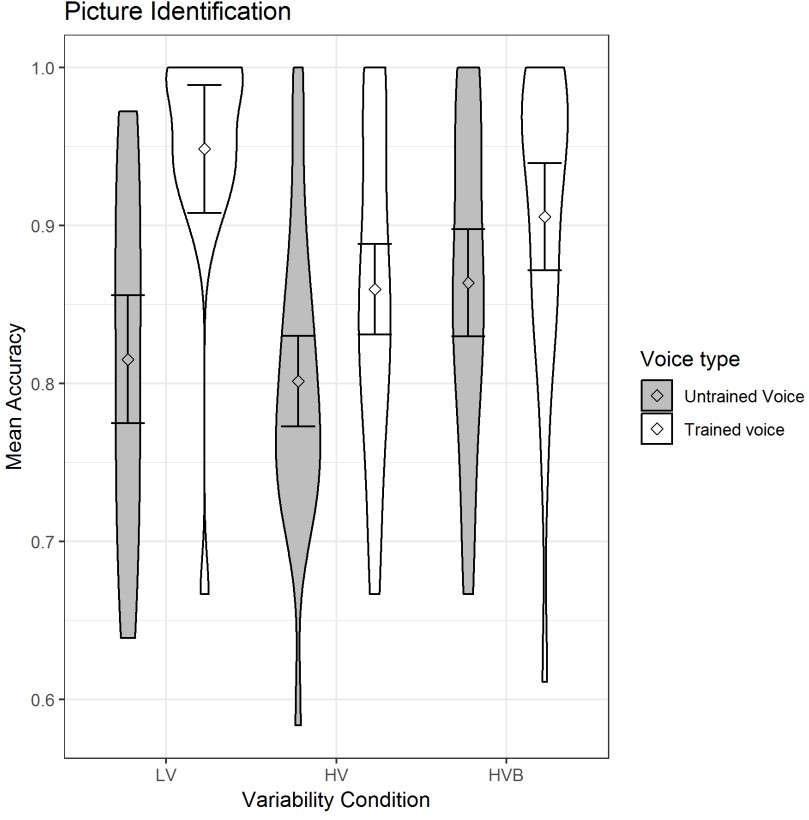

**Figure 6 Mean accuracy of Picture Identification for LV (low variability), HV (high variability) and HVB (high variability blocked) training groups for untrained voices and trained voices.** Error bars show 95% confidence intervals.

There was a main effect of *voice-novelty* ($\beta = 1.07$, SE = 0.16, $z = 6.53$, $p < 0.001$) reflecting higher performance in trials with trained voices. Although participants in the LV group performed better than those in the HV group ($\beta = -0.71$, SE = 0.32, $z = -2.23$, $p = 0.03$), there was no significant difference between the LV and the HVB group ($\beta = -0.14$, SE = 0.32, $z = -0.44$, $p = 0.66$) and there was a significant interaction between *voice-novelty* and both the LV-HV contrast ($\beta = -1.19$, SE = 0.35, $z = -3.43$, $p < 0.01$) and the LV-HVB contrast ($\beta = -1.11$, SE = 0.36, $z = -3.08$, $p < 0.01$). Breaking this down by *variability-condition*: for each condition there was significantly better performance with trained than untrained voices (LV: $\beta = 1.83$, SE = 0.29, $z = 6.42$, $p < 0.001$; HV: $\beta = 0.64$, SE = 0.23, $z = 2.86$, $p < 0.01$; HVB: $\beta = 0.73$, SE = 0.26, $z = 2.82$, $p < 0.01$), indicating greater ease with the familiar voice. Breaking down by *voice-novelty*: For the trained voice, performance was higher in the LV condition than in either the HV or HVB conditions, although this was only significant for the LV vs HV contrast (LV vs HV: $\beta = -1.30$, SE = 0.44, $z = -2.97$, $p < 0.01$; LV vs HVB: $\beta = -0.70$, SE = 0.45, $z = -1.55$, $p = 0.12$). Importantly, for untrained voices, neither of the contrasts between conditions was significant (LV vs HV: $\beta = -0.12$, SE = 0.26, $z = -0.45$, $p = 0.65$; LV vs HVB $\beta = 0.41$, SE = 0.27, $z = 1.51$, $p = 0.13$), indicating no evidence for greater generalisation following HV training.

## Production tests

### Coding and inter-rater reliability analyses

The same methods were used for both production tests. The files were combined into a single set, along with the 360 stimuli which were used in the experiment (and which were produced by native Mandarin speakers). The latter items were included in order to examine whether the raters were reliable. All stimuli were rated by two raters: Rater 1 was the first author and Rater 2 was recruited from the UCL MA Linguistics program and was naïve to the purposes of the experiment. Raters were presented with recordings in blocks in a random sequence (blind to test-type, condition, whether the stimulus was from pre-test or post-test and whether it was produced by a participant or was one of the experimental stimuli). For each item, raters were asked to (i) identify the tone, (ii) give a rating quantifying how native-like they thought the pronunciation was compared (one to seven with one as not recognisable and seven as native speaker level), and (iii) transcribe the pinyin (segmental pronunciation) produced by the participants.

If there was no sound or the tone was unrecognizable, the rater coded 0 when identifying the tone. Data from these trials were removed from the dataset before analyses were conducted. In addition, all of the data from one participant was removed from the analyses due to bad recording quality resulting from a technical error. In total, this resulted in 3.38% (359/10,620) of production trials being removed from analysis (*Word Repetition*: Pre-test 1.98% (84/4,248); Post-test 3.72% (158/4,248); *Picture Naming* 5.51% (117/2,124)). Three measurements were taken from the production tasks: mean accuracy of tone identification (Tone accuracy), mean tone rating (Tone rating) and mean accuracy of production in pinyin (derived by coding each production as correct (1 = the entire string is correct) or incorrect (0 = at least one error in the pinyin)). As a first test of rater reliability, performance with the native speaker stimuli was examined–these were near ceiling: Rater 1: Tone accuracy = 98%, Tone rating = 6.7, Pinyin accuracy = 80%; Rater 2: Tone accuracy = 87%, Tone rating = 6.5, Pinyin accuracy = 80%).

Furthermore, for the remaining data (i.e. the experimental data) inter-rater reliability was examined for all three measures for the two production tasks. For the binary measures (Tone accuracy and Pinyin accuracy), kappa statistics were calculated using the 'fmsb' package in R (*Cohen, 2014*). For the Word Repetition data, for Tone accuracy *kappa* = 0.39 ('fair agreement'), and for Pinyin accuracy *kappa* = 0.33 ('fair agreement'; *Landis & Koch, 1977*). For the Picture Naming test, for Tone accuracy *kappa* = 0.67 ('substantial agreement') and for Pinyin accuracy *kappa* = 0.53 ('moderate agreement'); For the Tone rating, the package 'irr' in R was used to assess the intra-class correlation (*McGraw & Wong, 1996*) based on an average-measures, two-way mixed-effects model. For Word Repetition, ICC = 0.22 and for Picture Identification ICC = 0.37; according to *Cicchetti (1994)*, values less than 0.40 are regarded as 'poor'. Given this, we do not include analyses with Tone Rating as the dependent variable (though these data are included in the data set https://osf.io/wdh8a/). All of the analyses presented in the sections 'Word Repetition' and 'Picture Naming' were based on Rater 2 (the naive rater).

## Word Repetition-Tone accuracy

**Figure 7** **Accuracy of Word Repetition for LV (low variability), HV (high variability) and HVB (high variability blocked) training groups in Pre- and Post-tests for trained and untrained items.** Error bars show 95% confidence intervals.

### Word repetition

*Tone accuracy*

The predicted variable was whether a correct response was given (1/0) on each trial (as identified by the coder). The predictors were *test-session* (pre-test, post-test), *variability-condition* (LV vs HV, LV vs HVB) and *item-novelty* (trained, untrained). The mean accuracy, split by test-session and training condition, is shown in Fig. 7.

At pre-test, there was no significant difference between the LV and the HV group ($\beta = 0.01$, SE = 0.18, $z = 0.06$, $p = 0.95$) nor between the LV and the HVB group ($\beta = 0.11$, SE = 0.18, $z = 0.64$, $p = 0.53$), suggesting the groups started at a similar level. There was also no difference between trained and untrained words at pre-test ($\beta = -0.02$, SE = 0.07, $z = -0.26$, $p = 0.80$).

Across the three groups, participants' performance increased significantly after training (*Mpre* = 0.71, *SDpre* = 0.09, *Mpost* = 0.79, *SDpost* = 0.09, $\beta = 0.40$, SE = 0.08, $z = 5.29$, $p < 0.001$). There was no significant difference in the improvement for trained and untrained items (*word-type* by *test-session* interaction: $\beta = 0.13$, SE = 0.10, $z = 1.22$ $p = 0.22$). Critically, the interactions between the variability contrasts and *test-session* were not significant (LV vs HV: $\beta = -0.10$, SE = 0.18, $z = -0.55$, $p = 0.58$; LV vs HVB: $\beta = -0.11$, SE = 0.18, $z = -0.62$, $p = 0.54$), and they were not qualified by any higher level interactions with *item-novelty* (LV vs HV: $\beta = 0.15$, SE = 0.25, $z = 0.61$, $p = 0.54$; LV vs HVB: $\beta = -0.31$, SE = 0.26, $z = -1.21$, $p = 0.23$). This suggests there is no

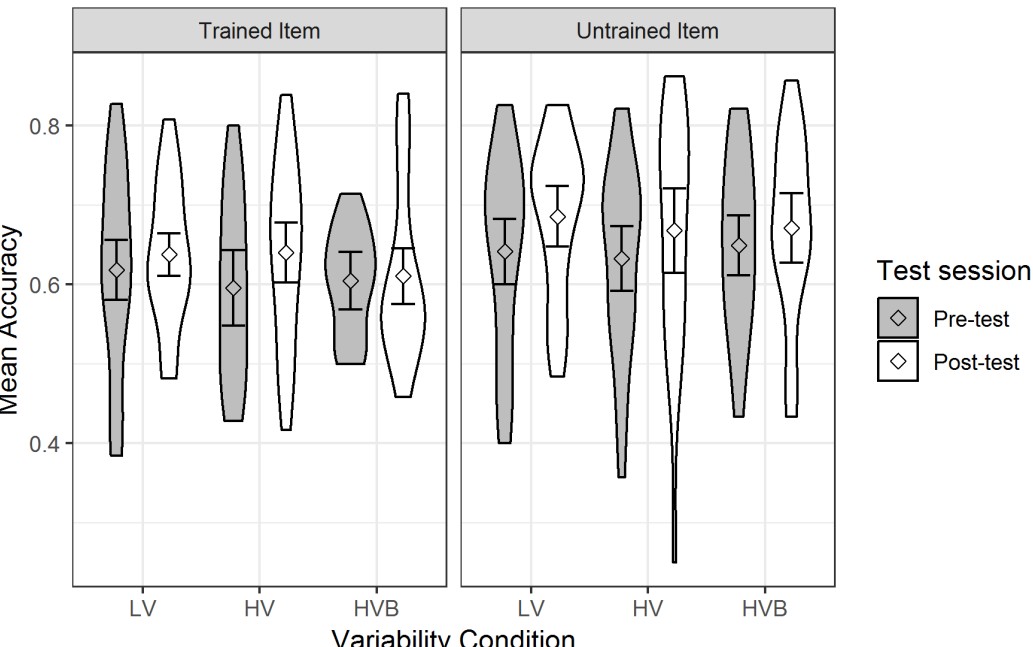

**Figure 8** Mean pinyin accuracy of Word Repetition for LV (low variability), HV (high variability) and HVB (high variability blocked) training groups in Pre- and Post-tests for trained and untrained items. Error bars show 95% confidence intervals.

evidence that participants' improvement in their production of tones was affected by their *variability-condition*, or that this differed for *trained* vs *untrained* items.

*Pinyin accuracy*

The predicted variable was whether the participants produced the correct string of phonemes (1/0) in each trial (as determined by Rater 2). The predictors were *test-session* (pre-test, post-test), *variability-condition* (LV vs HV, LV vs HVB) and *item-novelty* (trained, untrained). Mean pinyin accuracy is displayed in Fig. 8.

At pre-test, there was no significant difference between the LV and the HV group ($\beta = -0.01$, SE $= 0.11$, $z = -0.11$, $p = 0.91$) nor between the LV and the HVB group ($\beta = -0.03$, SE $= 0.11$, $z = -0.24$, $p = 0.81$), suggesting that the groups started at a similar level. However, participants did better on untrained words than trained words at pre-test ($\beta = 0.21$, SE $= 0.07$, $z = 3.11$, $p < 0.01$), suggesting potential accidental differences in these items. Participants showed significant improvement after training ($Mpre = 0.54$, $SDpre = 0.09$, $Mpost = 0.58$, $SDpost = 0.19$, $\beta = 0.15$, SE $= 0.05$, $z = 3.38$, $p < 0.01$). However, there was no evidence that different variability conditions resulted in different amounts of improvement (*test-session* by LV vs HV: $\beta = 0.05$, SE $= 0.11$, $z = 0.46$, $p = 0.65$; *test-session* by LV vs HVB: $\beta = -0.12$, SE $= 0.11$, $z = -1.08$, $p = 0.28$) or any interaction between *variability condition*, *test-session* and *item-novelty* (LV vs HV: $\beta = 0.11$, SE $= 0.22$, $z = 0.51$, $p = 0.61$; LV vs HVB: $\beta = -0.14$, SE $= 0.22$, $z = -0.64$, $p = 0.52$). This suggests

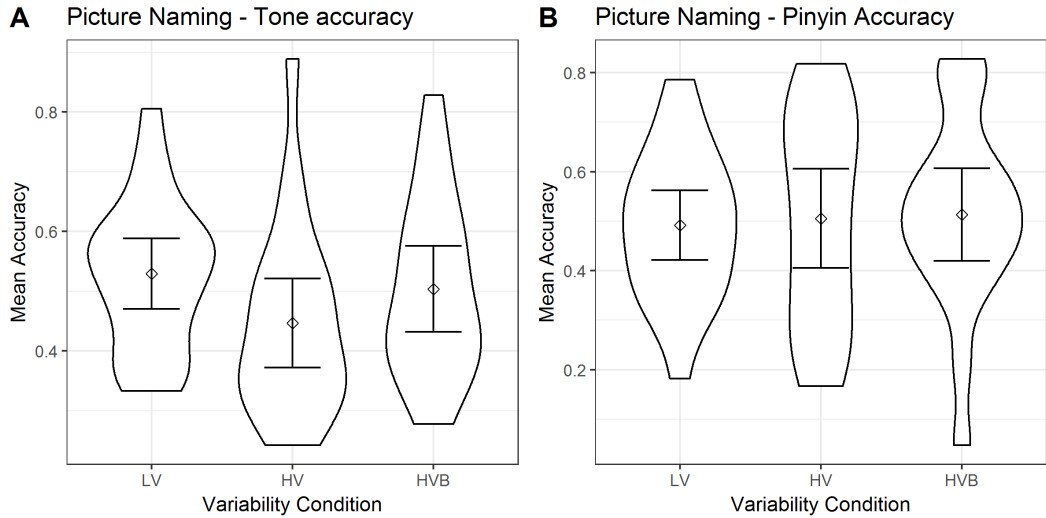

**Figure 9 Tone accuracy and Pinyin accuracy of Picture Naming for LV (low variability), HV (high variability) and HVB (high variability blocked) training groups. Error bars show 95% confidence intervals.** (A) Mean accuracy of Picture Naming, tone accuracy measure. (B) Mean accuracy of Picture Naming, pinyin accuracy measure.               

there is no evidence that participants' improvement in pinyin accuracy was affected by their *variability-condition*, or that this differed for *trained* vs *untrained* items.

### Picture naming

*Tone accuracy*

The predicted variable was whether a correct response was given (1/0) on each trial (as identified by the coder). There was only one predictor, *variability-condition* (LV vs HV, LV vs HVB) for both models. The descriptive statistics are displayed in Fig. 9.

Participants in the LV group showed no significant difference compared with the HV group ($\beta = -0.34$ SE = 0.19, $z = -1.81$, $p = 0.07$) and the HVB group ($\beta = -0.10$, SE = 0.19, $z = -0.52$, $p = 0.61$. This suggests there is no evidence that participants' ability to produce the tones accurately differed according to their *variability-condition*.

*Pinyin accuracy*

The predicted variable was whether the participants produced the correct string of phonemes (1/0) in each trial and there was a single predictor *variability-condition* (LV vs HV, LV vs HVB). For both models there was no significant difference between variability conditions (LV vs HV: $\beta = 0.09$, SE = 0.23, $z = 0.41$, $p = 0.68$; LV vs HVB: $\beta = 0.12$, SE = 0.23, $z = 0.51$, $p = 0.61$). This suggests there is no evidence that participants' pinyin accuracy differed according to their *variability-condition*.

### Analyses with individual aptitude

In order to look at the effect of learner aptitude and the interaction between this factor and variability condition, we first calculated the mean accuracy at pre-test on the Pitch Contour Perception Test for each participant. This score (scaled by a factor of 10, so that each one unit increase in aptitude corresponded to a 10% higher performance in the Pitch Contour

**Table 4 Statistics obtained when adding in participant aptitude (as measured by performance on the Pitch Contour Perception Test task at pre-test) into the models predicting performance on the test and training tasks.** Statistics marked in bold are significant (0.05) results.

| Data set | Coefficient name | Statistics |
|---|---|---|
| Word repetition: Tone accuracy (Pre/post) | **Aptitude** | **$\beta = 0.07$, SE $= 0.03$, $z = 2.35$, $p = 0.019$** |
| | Aptitude by *Test-Session* | $\beta = 0.03$, SE $= 0.04$, $z = 0.72$, $p = 0.473$ |
| | Aptitude by LV-HV Contrast by *Test-Session* | $\beta = 0.05$, SE $= 0.11$, $z = 0.47$, $p = 0.639$ |
| | Aptitude by LV-HVB Contrast by *Test-Session* | $\beta = 0.13$, SE $= 0.10$, $z = 1.35$, $p = 0.176$ |
| | Aptitude by LV-HV Contrast by *Test-Session* by *Item-Novelty* | $\beta = -0.14$, SE $= 0.15$, $z = -0.97$, $p = 0.334$ |
| | Aptitude by LV-HVB Contrast by *Test-Session* by *Item-Novelty* | $\beta = 0.07$, SE $= 0.13$, $z = 0.50$, $p = 0.61$ |
| Three interval oddity (Pre/post) | **Aptitude** | **$\beta = 0.07$, SE $= 0.03$, $z = 2.19$, $p = 0.029$** |
| | Aptitude by *Test-Session* | $\beta = 0.01$, SE $= 0.23$, $z = 0.31$, $p = 0.757$ |
| | Aptitude by LV-HV Contrast by *Test-Session* | $\beta = 0.05$, SE $= 0.07$, $z = 0.77$, $p = 0.443$ |
| | Aptitude by LV-HVB Contrast by *Test-Session* | $\beta = 0.05$, SE $= 0.06$, $z = 0.83$, $p = 0.410$ |
| | Aptitude by LV-HV Contrast by *Test-Session* by *Item-Novelty* | $\beta = -0.12$, SE $= 0.13$, $z = -0.94$, $p = 0.346$ |
| | Aptitude by LV-HVB Contrast by *Test-Session* by *Item-Novelty* | $\beta = 0.06$, SE $= 0.11$, $z = 0.52$, $p = 0.604$ |
| Training | **Aptitude** | **$\beta = 0.13$, SE $= 0.048$, $z = 2.70$, $p = 0.007$** |
| | Aptitude by LV-HV Contrast | $\beta = -0.04$, SE $= 0.11$, $z = -0.332$, $p = 0.740$ |
| | Aptitude by LV-HVB Contrast | $\beta = 0.03$, SE $= 0.10$, $z = 0.26$, $p = 0.795$ |
| Picture identification (Post only) | **Aptitude** | **$\beta = 1.48$, SE $= 0.08$, $z = 1.96$, $p = 0.050$** |
| | Aptitude by Voice Novelty | $\beta = -0.03$, SE $= 0.07$, $z = -0.33$, $p = 0.745$ |
| | Aptitude by LV-HV Contrast | $\beta = -0.02$, SE $= 0.19$, $z = -0.12$, $p = 0.901$ |
| | Aptitude by LV-HVB Contrast | $\beta = 0.01$, SE $= 0.17$, $z = 0.09$, $p = 0.932$ |
| | Aptitude by LV-HV Contrast by *Voice-Novelty* | $\beta = 0.35$, SE $= 0.21$, $z = 1.63$, $p = 0.103$ |
| | Aptitude by LV-HVB Contrast by *Voice-Novelty* | $\beta = -0.11$, SE $= 0.19$, $z = -0.58$, $p = 0.566$ |
| Picture naming: tone accuracy | Aptitude | $\beta = 0.08$, SE $= 0.04$, $z = 1.89$, $p = 0.059$ |
| | Aptitude by LV-HV Contrast | $\beta = -0.09$, SE $= 0.11$, $z = -0.84$, $p = 0.402$ |
| | Aptitude by LV-HVB Contrast | $\beta = 0.12$, SE $= 0.10$, $z = 1.22$, $p = 0.224$ |

Perception test) was centred and used as a continuous predictor (*aptitude*) and added to each of the models reported above. In addition, we added the interaction between this factor and key experimental factors (see Table 4). Based on *Perrachione et al. (2011)* and *Sadakata & McQueen (2014)*, for our measures of tone-learning, HV should benefit high aptitude participants only, while LV would benefit low aptitude participants only. In our design, we used a continuous measure of individual ability rather than a binary division of high and LV. We therefore predicted a stronger positive correlation between *aptitude* and amount of learning in the HV condition than in the LV condition. In the tests administered only post training (i.e. Picture Identification and Picture Naming) this would show up as an interaction between aptitude and condition. In the models for the pre- and post-test data (i.e. Three Interval Oddity and Word Repetition) this would show up as a three-way interaction between *condition, test-session* and *aptitude*. We also looked at the interactions between these factors and *voice-novelty* (Picture Identification) and *item-novelty* (Three Interval Oddity and Word Repetition). Note that there are no clear directional hypotheses here: *Perrachione et al. (2011)* found the interaction in a test with untrained voices and trained items, and *Sadakata & McQueen (2014)* found

[4] This was the case even if we split the data into two models, as we did in the Section 'Training'.

the interaction in a test with trained voices and trained items. For training, in principal both the two-way interaction of *aptitude* by *condition* and the three-way interaction of *aptitude* by *condition* by *training-session* are of interest. However, it was not possible to fit a converging model containing the three-way factor[4].

Each model reported in Table 4 contained all the fixed effects included in the original models in addition to the fixed effects listed in the table (note that to avoid convergence issues due to over complex models, we did *not* attempt to include the complete set of interactions for every combination of experimental variables with aptitude—only those for which we had predictions). We attempted to have full random effects structure for these fixed effects however in some cases we had to remove correlations between slopes due to problems with convergence and for one of the models with the training data we had to remove the random slope for training session). Note that we don't include models for the pinyin measures, since our measure of aptitude is relevant to tone learning only. For each of the new models we first confirmed that adding in the new effects and interactions with the individual measures did not change any of the previously reported patterns of significance for the experimental effects (see script https://osf.io/wdh8a/) for full models.

The results are shown in Table 4. *Aptitude* is a positive predictor of performance in each of the tests and in training, with *p*-values significant or marginal in each case. However there was no interaction between *aptitude* and any other factor. Thus, there was no evidence that this measure of aptitude correlated with participants ability to benefit from training (no interaction with *test-session*), nor—critically for our hypothesis—did this differ by training condition (no interaction with *condition* or with *test-session* by *condition*).

Although the analyses use a continuous measure of Pitch Contour Perception Test, for the purposes of visualisation, Fig. 10 (Three Interval Oddity task and Training task), Fig. 11 (Picture Naming and Picture Identification) and Fig. 12 (Word Repetition) use the mean accuracy for participants split into aptitude groups using a median split based on their Pitch Contour Perception Test score.

In sum, participants with higher aptitude measures were better at the tasks, but there is no evidence either that this affected their improvement due to training, or, critically, their ability to benefit from the different variability exposure sets.

## Bayes factor analyses

In the analyses reported above, we did not find evidence—in any of our tests—for either of two key hypotheses: (1) the hypothesis that training with multiple speakers leads to greater generalisation to new speakers than training with a single speaker *or* (2) the hypothesis that there is an interaction between the variability of the training materials and participant aptitude, such that higher aptitude participants benefit more from training with multiple speakers while lower aptitude participants benefit more from training with a single speaker. However, there is a difficulty in interpreting these null results since a non-significant result ($p > 0.05$) does *not* tell us whether we have evidence for the null, as opposed to no evidence for any conclusion at all, or even evidence against the null.

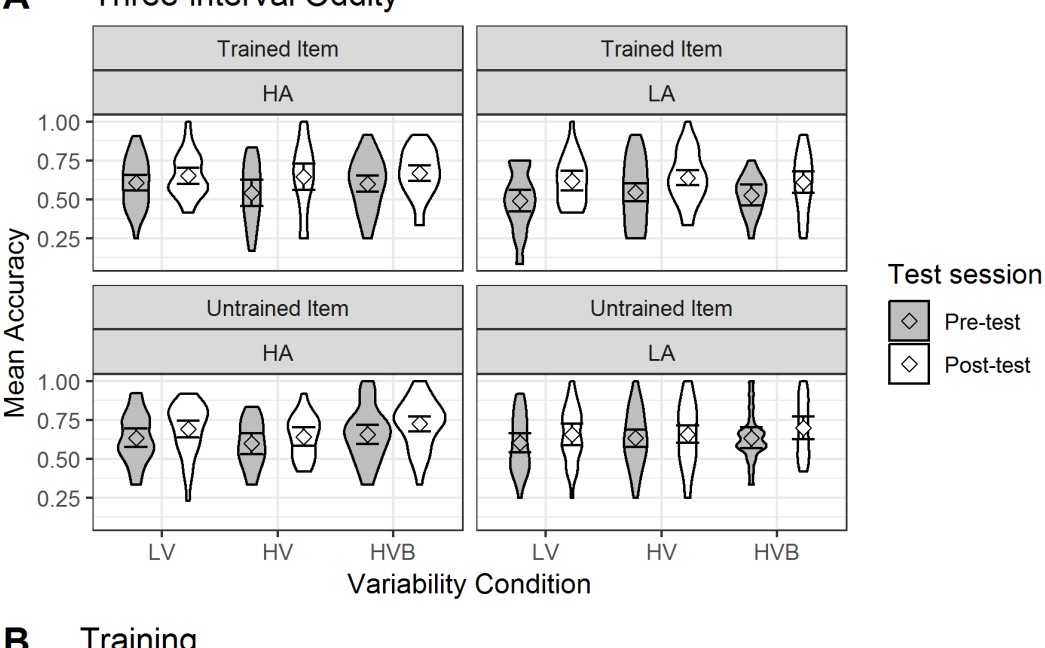

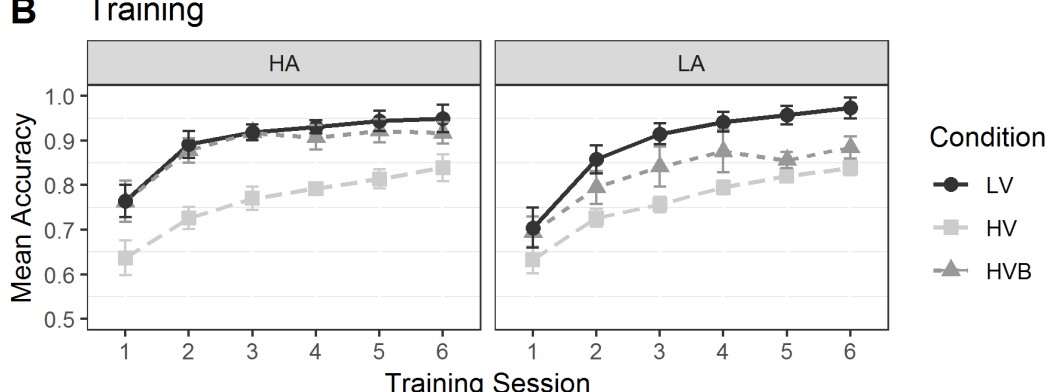

**Figure 10 Accuracy in Three Interval Oddity and Training for LV (low variability), HV (high variability) and HVB (high variability blocked) training groups.** Error bars show 95% confidence interval. (A)Mean accuracy of Three Interval Oddity, split by high (HA) vs low (LA) aptitude in the Pitch Contour Perception Test (B) Mean accuracy of Training, split by high (HA) vs low (LA) aptitude in the Pitch Contour Perception Test.

Thus, we should not reduce our confidence in either of our hypotheses on the basis of the null results reported above (despite the fact that reducing confidence in a theory following non-significant results is common practice)—see *Dienes (2014)* for discussion. An alternative statistic is a Bayes Factor, which are used to assess the strength of evidence for one theory (H1) over another (the null hypothesis). We therefore supplemented the analyses above by computing Bayes factors for contrasts relating to these two key hypotheses. These are reported in the sections 'H1: Greater generalization—to Novel Voices and in Production—in the Multiple Speaker Conditions (HV and HVB) than in the LV Condition' and H1: There is an Interaction Between an Individual's Tone-Aptitude and Variability-Condition, Such That Participants with Greater Tone-Aptitude Show Greater Performance Following the Multiple Speaker Conditions (HV and HVB) and

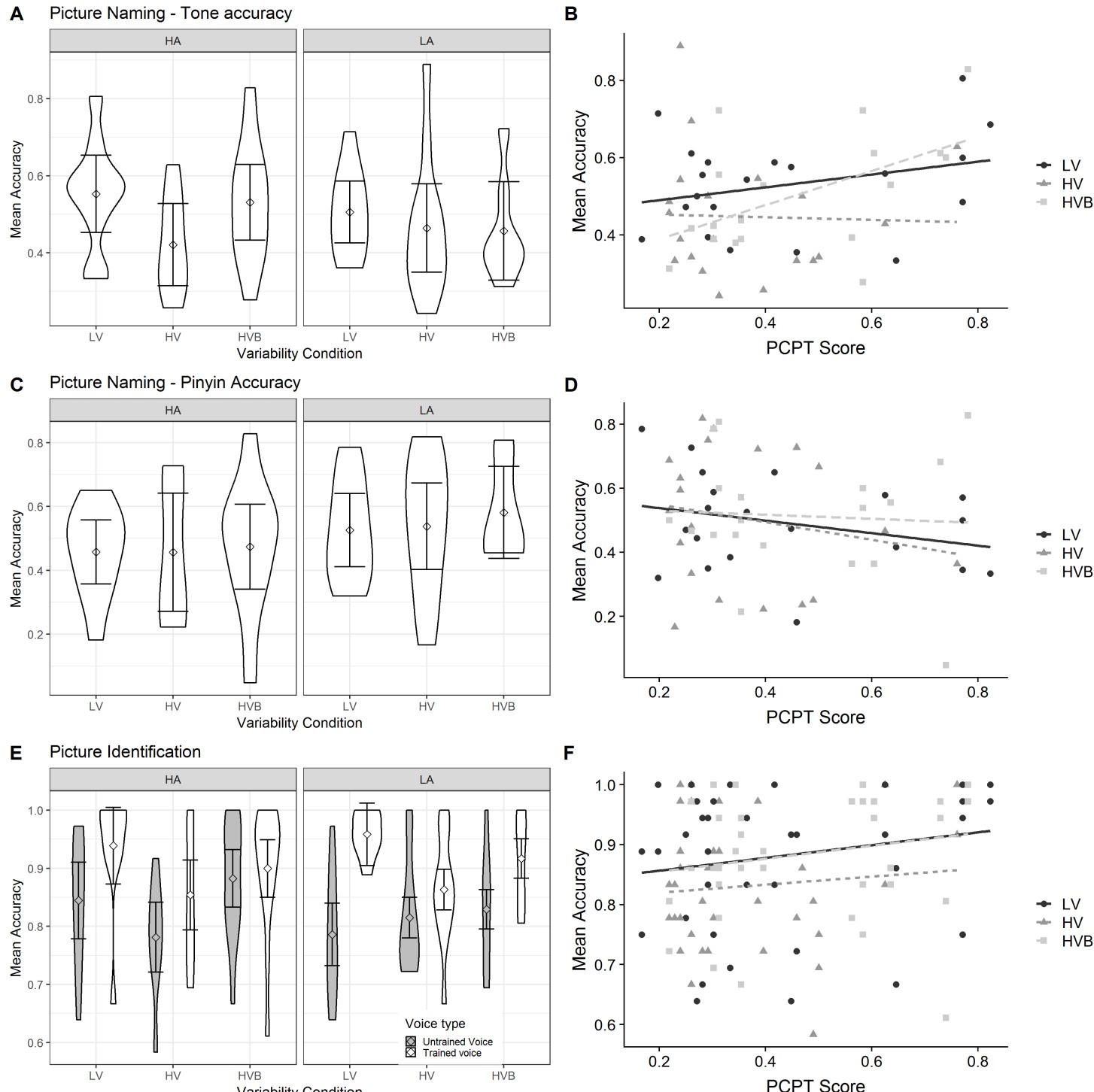

**Figure 11 Accuracy in Picture Naming and Picture Identification for LV, HV and HVB training groups, split by high (HA) vs low (LA) aptitude in the Pitch Contour Perception Test.** Error bars show 95% confidence interval. (A) Mean accuracy of Picture Naming tone accuracy measure (B) Scatter plot contrasting Mean accuracy of Picture Naming tone accuracy measure and corresponding aptitude measure from Picture Contour Perception Test (C) Mean accuracy of Picture Naming Pinyin accuracy measure (D) Scatter plot contrasting Mean accuracy of Picture Naming Pinyin accuracy measure and corresponding aptitude measure from Picture Contour Perception Test (E) Mean accuracy of Picture Identification (F) Scatter plot contrasting Mean accuracy of Picture Identification and corresponding aptitude measure from Picture Contour Perception Test.

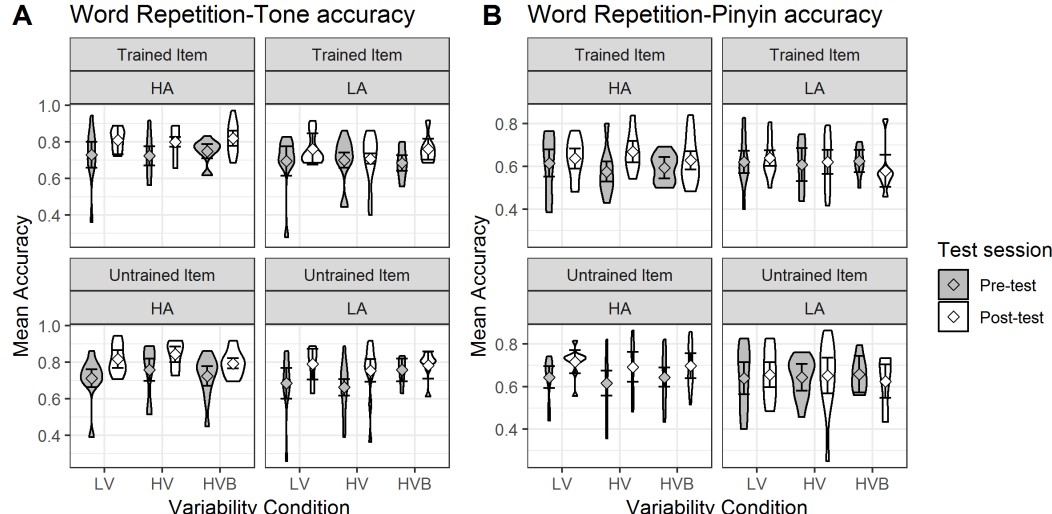

**Figure 12 Accuracy in Word Repetition for LV, HV and HVB training groups, split by high (HA) vs low (LA) aptitude in the Pitch Contour Perception Test.** Error bars show 95% confidence intervals. (A) Mean accuracy of Word Repetition tone accuracy measure (B) Mean accuracy of Word Repetition Pinyin accuracy measure.

Those with Lesser Tone Aptitude Show Greater Performance in the Single Speaker Condition (LV)' below.

### H1: Greater generalisation—to novel voices and in production—in the multiple speaker conditions (HV and HVB) than in the low variability condition (LV)

We aimed to compute Bayes Factors comparing this hypothesis to the null for each of our data sets. To have maximum evidence, we pool the HV and HVB conditions and contrast this with the LV condition. For the post-tests we are interested in the evidence for a main effect of this contrast. For the pre-post tests, we are interested in the interaction between this contrast and session. To further maximise evidence, for the Three Interval Oddity test and Word Repetition tests we look at trained and untrained items combined (since *both* types of item involve generalisation to an untrained voice and thus should benefit from HV training), however in the Picture Identification test we excluded trained *voice* test items, since the benefit of HV training was not predicted for these items. For the production measures, we are interested in whether there is a HV benefit for our tone learning measure and our pinyin measure (the latter given that *Barcroft & Sommers (2014)*, found a benefit of multi-speaker training in their vocabulary recall task).

We computed Bayes factors following *Dienes (2014)* and *Dienes, Coulton & Heather (2018)*. To compute a Bayes factor (*B*) it is necessary to have both a model of the data and a model of H1. The model of the data is an estimate of the mean difference for the contrast in question, and of the standard error. Here, we get these estimates by running logistic mixed models and taking the betas and standard errors for the relevant coefficients (note that this allows us to meet normality assumptions by continuing to work within log-odds space). The models we ran here were similar to the previous

analyses but with variability-condition coded as a centred contrast between LV and the HV+HVB conditions, and other factors combined/excluded as described in the previous paragraphs. The full set of models is in https://osf.io/wdh8a/.

We model H1 using a half-normal distribution with a mode of 0 and a standard deviation $x$ which is set to be a rough estimate of the predicted difference for this contrast. This allows for possible effects between 0 and twice the predicted effect, with values closer to 0 being more likely (*Dienes, 2014*).

In the absence of any prior data using sufficiently similar materials, and since we did not wish to use unprincipled default values, we estimated $x$ for each contrast using the scale and/or values from elsewhere in the data (see *Dienes, 2014*, *2015* for a related approach). Specifically, for each of the cases where we predicted a main effect (Picture Identification and Picture Naming), we set $x$ as the difference between the grand mean (the Intercept—since we use a centred coding) and an estimate of minimal possible performance on the task. The logic is as follows[5]: The *maximum* difference between conditions is seen if LV participants show baseline performance and HV participants show performance greater than baseline. In this case, if performance on this test is $p$ (so the grand mean is $\bar{p}$) and the baseline is $b$, the difference in $p$ between the two conditions will be equal to: $2(\bar{p} - b)$. This gives us an estimate of the *maximum* value of $x$; since we are using a half normal distribution with a mean of zero, we assume the maximum value is equal to approximately 2SD, so we can set our estimate $x$ of the standard deviation to be equal to *half* of this value (i.e. $x = \bar{p} - b$). Baseline performance depends on the task: for the 2AFC Picture Identification task it is chance (50% = 0 in log odds space); for the Picture Naming, tone measure, we assume a ¼ chance of identifying the correct one (25% = −1.099 in log odds space); for Picture Naming, Pinyin measure, there is no chance and we therefore took minimal performance as making one correct response in the test[6] (i.e. 1/72 = −4.263 in log odds space). For the cases where we are estimating an interaction between *test-session* and *variability-condition* we set $x$ as equal to the mean increase in performance from *pre-* and *post*-test across conditions (main effect of *test-session*). The logic is as follows: the *maximum* difference is seen if LV participants show no effect of *test-session* (no improvement) and HV participants show a positive effect of *test-session*. In this case, if the mean effect of *test-session* is $\bar{t}$, the difference in $t$ between the two conditions will be equal to $2\bar{t}$. Again, we can set our estimate of $x$ to be half this value (i.e., $x = \bar{t}$).

We interpret BFs using the following conventions: $B < 1/3$ indicates substantial evidence for the null, $B > 3$ indicates substantial evidence for H1, values between 1/3 and 3 indicate that the data collected do not sensitively distinguish H0 from H1 (*Jeffreys, 1998*; *Dienes, 2008*). Since there is subjectivity in how the values for H1 are determined, we indicate the robustness of Bayesian conclusions by reporting a robustness region for each $B$, which gives the range of values of the scale factor $x$ that qualitatively support the same conclusion (i.e. evidence as supporting H0, or as supporting H1, or there not being much evidence at all). Note that for evidence for H0, the maximum $x$ is always infinity. The results are reported in Table 5. It can be seen we have substantial or strong evidence for the null for every test except for the Word Repetition test for the Pinyin accuracy measure, where

[5] Further details of the logic of these computations is spelt out in the script available at https://osf.io/wdh8a/.

[6] Note that we cannot compute log-odds of 0.

**Table 5 Bayes Factor results testing the hypothesis that there is greater generalisation following either of the high variability training conditions than the low variability condition.**

| Contrast | Mean difference | Stand. Error | H1 estimate $x$ | Bayes factor ($B$) | Robustness region |
|---|---|---|---|---|---|
| Picture ID (Novel voice only) HV+ HVB > LV | 0.13 | 0.228 | 1.71 | 0.219 | 1.11 : ∞ |
| Picture naming, (Tone accuracy) HV+ HVB > LV | −0.225 | 0.168 | 1.076 | 0.067 | 0.202 : ∞ |
| Picture naming (Pinyin Accuracy) HV+ HVB > LV | 0.104 | 0.196 | 4.05 | 0.08 | 0.101 : ∞ |
| Word repetition (Tone accuracy) *test-session* by HV+ HVB > LV | −0.108 | 0.157 | 0.395 | 0.239 | 0.303 : ∞ |
| Word repetition (Pinyin accuracy) *test-session* byHV+ HVB > LV | 0.095 | −0.034 | 0.152 | 0.421 | 0 : 0.202 |
| Three interval oddity*test-session* by HV+ HVB > LV | −0.001 | 0.1 | 0.31 | 0.303 | 0.303 : ∞ |

the evidence is ambiguous, and that the robustness regions indicate that we would continue to have evidence for the null even with smaller estimates of the scale factor $x$.

### H1: There is an interaction between an individual's tone-aptitude and variability-condition, such that participants with greater tone-aptitude show greater performance following the multiple speaker conditions (HV and HVB) and those with lesser tone aptitude show greater performance in the single speaker condition (LV)

We aimed to compute Bayes Factors comparing this hypothesis to the null for each of our data sets. We take the same approach as above except that we also compute Bayes factors for Training data, and for the Picture Identification test we look at both trained voice and untrained voice data—pooling the two in order to maximise available evidence. This is because this interaction has been reported with trained items (*Sadakata & McQueen, 2013*) as well as untrained items (*Perrachione et al., 2011*). We again combine the HV and HVB conditions except for training where we look at the LV vs HV and LV vs HVB contrasts separately, since we have seen in our previous analyses that HV and HVB are quite different (HVB participants show higher performance).We again combine the evidence from trained and untrained *items* in the pre-post tests. For the post-session only tests, we are interested in the evidence for an interaction between the *variability-condition* contrast and *aptitude*. For the tests which appeared both pre- and post-training, we are interested in the interaction between the *variability-condition* contrast, *aptitude* and *test-session*. For training we look at the evidence for an interaction between each *variability-condition* contrast and *aptitude* (a more complex model containing the interaction with training-session did not converge). As in our frequentist analyses of aptitude, for the production measures—Word Repetition and Picture Naming—we do *not* look at the pinyin measures since our aptitude measure is relevant only to tone learning.

We computed Bayes factors following the same procedure as in the section 'H1: Greater Generalization—to Novel Voices and in Production—in the Multiple Speaker Conditions (HV and HVB) than in the LV Condition' and again derived our estimates of the scale factor $x$—the difference predicted under H1—using the scale and/or values from elsewhere in the data. Specifically, for each of the cases where we predicted a two-way interaction between *variability-condition* and *aptitude* we set $x$ as equal to the mean

**Table 6 Bayes Factor results testing the hypothesis that there is an interaction between aptitude and variability-condition greater generalisation following either of the high variability training conditions than the low variability condition.**

| Contrast | Mean difference | Stand. Error | H1 estimate $x$ | Bayes factor ($B$) | Robustness region |
|---|---|---|---|---|---|
| ID, (Tone accuracy) *aptitude* by HV+ HVB > LV | 0.006 | 0.127 | 0.171 | 0.617 | 0 : 0.354 |
| Picture naming, (Tone accuracy) *aptitude* by HV+ HVB > LV | 0.042 | 0.083 | 0.099 | 0.904 | 0 : 0.354 |
| Three interval oddity (Tone accuracy) *aptitude* by *test-session* by HV+ HVB > LV | 0.048 | 0.05 | 0.345 | 0.371 | 0 : 0.354 |
| Word Repetition (Tone accuracy) *aptitude* by *test-session* by HV+ HVB > LV | 0.091 | 0.082 | 0.379 | 0.654 | 0 : 0.758 |
| Training *aptitude* by HV > LV | −0.037 | 0.119 | 0.129 | 0.572 | 0 : 0.253 |
| Training *aptitude* by HVB > LV | 0.026 | 0.101 | 0.129 | 0.732 | 0 : 0.354 |

[7] An alternative which would be more equivalent to the other BF analyses would be to inform the effect using the value of the two-way interaction of aptitude: test-session. We do not do this since we did not find an effect of this two-way interaction in either data set.

effect of *aptitude* across conditions (main effect of *aptitude*)[7]. The logic is as follows: The *maximum* difference is seen if LV participants show no effect of *aptitude* and the HV participants show a positive effect of *aptitude* (note that a negative effect of aptitude is not expected in any condition). In this case, if the mean effect of *aptitude* is $\bar{a}$, the difference in $a$ between the two conditions will be equal to $2\bar{a}$. Again, we can set our estimate of $x$—the SD of the half normal—to be half this maximum value, that is, $x = \bar{a}$. For the cases where we are interested in the three-way interaction between aptitude, test-condition and test-session, we based our estimate on half the difference between the maximal effect of aptitude *(maxA—taken from the scale)* and their actual aptitude score at pre-test *(baselineA—taken from the data)*. The logic is as follows: The maximal effect of the interaction would be seen if participants in the LV condition showed the same baseline effect of aptitude at *pre-test* and at *post-test (ba)*, whereas participants in the HV condition showed maximal improvement at post-test *(maxa)*. In this case, the interaction between aptitude and session for the *HV* group would be equal to: *maxa—ba*. Again, we can set our estimate of $x$—the SD of the half normal—to be half this maximum value, that is, $x = \frac{ma-ba}{2}$.

The maximum effect of aptitude was computed from the scale and the length of the aptitude predictor. Specifically, we assumed that the maximal effect of aptitude would be obtained if participants with maximal aptitude were at ceiling (71/72 correct—log odds 4.263) and those with minimal aptitude were at chance (25% in Word Repetition, Tone Accuracy, log odds= 1.099; 33.33% in Three Interval Oddity, log odds = 0.693). We divided this range by the length of the aptitude predictor to obtain a measure of a one-step change in aptitude.

The results are summarised in Table 6. It can be seen that although there is more evidence for the null than H1 in each case (i.e. BF < 1) we do *not* have substantial evidence for the null over H1 in any case. Thus, we cannot draw any inferences about the interaction from this data. Note that, in most cases, the robustness regions indicate that even if the scale factor $x$ was twice as large, that is, corresponding to the *maximum* value we might expect, the $B$ would be ambiguous.

## DISCUSSION

The current study investigated the effect of different types of phonetic training on English speakers' learning of novel Mandarin words and tones. To our knowledge, this is the

first study to train naive participants on all four Mandarin tones, using real language stimuli embedded in a word learning task. Learning was examined using a range of perception and production tasks. Following previous literature, we compared three training conditions: LV (single speaker), HV (four speakers, presented intermixed) and HV blocked (four speakers, presented in blocks). We also administered tests designed to tap individual aptitude in the perception of pitch contrasts, adapted from the previous literature. The results indicated that participants' performance increased during training and that training also led to improved performance on pre- to post-tests of discrimination and production, with evidence of generalisation to untrained voices and items. Participants also showed some ability to recall trained words—including their tones—in a picture naming task administered at post-test. However, the only place where we saw any effect of the variability manipulation was in the training task (and with trained items in the picture identification task, which was highly similar to training), where the *low* variability group outperformed both of the HV groups. Critically, we found no evidence in any of our tests that HV input benefitted learning or generalisation, nor did we find any evidence of an interaction between individual aptitude and the ability to benefit from HV training. In the following discussion, we first consider the findings from each task in turn before turning to a more general discussion of our findings in relation to the predicted benefit of HV input.

## Tests of individual aptitude

In the current work, we conducted two tests with the purpose of capturing individual aptitude: The Pitch Contrast Perception Test (following *Perrachione et al., 2011*) and the Categorisation of Synthesised Tonal Continua (following *Sadakata & McQueen, 2014*). Although our goal was to measure participants' baseline aptitude, the tests were conducted both at pre- and post-test, following Sadakata and McQueen, who did not find differences from pre- to post-tests with their categorisation measure, and who used combined data from pre- and post-test to compute participants slopes. Unfortunately, the performance of our own participants suggested that the Categorisation of Synthesised Tonal Continua test was not a good test of aptitude, with the majority of participants failing to meet the slope threshold used in Sadakata and McQueen, and most being unable to consistently categorise the end points of the continua. It is unclear why our results differ from the previous study (we aimed to follow their procedures), but this meant that we were unable to use this as an aptitude measure in our later analyses. The scores on the Pitch Contrast Perception Test alone therefore served as our measure of individual aptitude. Interestingly, preliminary analyses (section 'The Pitch Contour Perception Test') demonstrated that performance in this test improved from pre- to post-training. This suggests that this measure is not a 'pure' measure of individual differences since it also appears to be affected by experience. Given this, we only used participants' scores on this test from *pre-test* as the measure of aptitude in subsequent analyses.

## Performance in training

The training task employed in this study was a 2AFC task, where participants had to identify the correct meaning of a Mandarin word based on its tone. The results from

training indicate that participants performed better in the single speaker LV training than in either the multiple speaker HV or HVB groups. This difference was present from the first session for the LV-HV contrast, and from the second session for the LV-HVB contrast (i.e. the first session where the two conditions differ), and increased over time for both contrasts. Greater difficulty with multiple speaker input is in line with the findings of *Perrachione et al. (2011)*, although the differences did not emerge so rapidly in that study, possibly due to there being fewer trials per session. Intuitively, repeated exposure to the single speaker in the LV condition allows for greater adaptation to speaker specific cues, whereas in the HV conditions participants have to adapt to multiple speakers. This is particularly difficult in the unblocked HV condition, where trial-by-trial adaptation is needed, which is effortful for participants (*Magnuson & Nusbaum, 2007*). Importantly, however, for all three groups, their performance gradually increased over each session. In combination with the fact that their performance on the other tasks increased after training, this indicates that the training task and materials were effective. We also explored the role of learner aptitude in this task (as measured by performance on the Pitch Contour Perception Test at pre-test) and whether this influenced participant's performance differently in the different variability conditions. Overall, aptitude was found to be a significant predictor of performance during training. However, there was no evidence for an interaction with training condition, although our Bayes Factor Analyses suggests that the data here are inconclusive. We return to this finding in the section 'The Role of High Variability Materials in Training and Generalisation' below.

## Perception tests

We included two perceptual tasks which tapped learning and generalisation due to training: A *Picture Identification* task administered at post-test and a *Three Interval Oddity* task administered at both pre- and post-test. The *Picture Identification* task was a version of the training task without feedback, and is the most similar to the tests used by *Perrachione et al. (2011)*, and *Sadakata & McQueen (2014)*. We used this test to look at learning of the trained stimuli, comparing trained and untrained voices. The three interval oddity task had not been used in the previous studies, but allowed us to use a pre-/post-test design, and also to look at participants' performance with *untrained items*. These tests provided evidence that participants improved in their perception of tones following training: They were above chance in using the tone to identify the correct picture in the picture identification task at post-test, and they improved in their ability to discriminate tones in the three interval oddity task (59% performance prior to training, 66% post training). There was also evidence of generalisation across both voices and items: Participants were above chance in identifying the correct pictures even with an untrained voice (although they did show significantly weaker performance than with the trained voice) and they improved in their ability to discriminate the between minimal pair items in the three interval oddity task, even for untrained items.

Our key questions concerned the role of variability in training. First, we were interested in whether there was evidence that exposure to multiple voices during training led to greater ability to generalise across voices at test—that is, greater performance with novel

voices in the HV conditions than in the LV condition. We did not see this. In fact, the only effect of variability in this data was a *low* variability benefit, which we saw in the Picture Identification task for the trained-voice items (seen in the contrast between LV and HV conditions). This mirrors what we saw in training and reflects the greater exposure to this particular speaker in the LV training. However, in the tests tapping generalisation to a novel speaker—that is, in untrained voice trials in the Picture Identification task, and with all of the test-items in the Three Interval Oddity task, there was no difference between variability-training conditions. Bayes factor analyses indicate that in both cases, there was substantial evidence for the null.

The second hypothesis was that there would be an interaction between learner aptitude (as measured by the Pitch Contour Perception Test at pre-test) and variability training condition, such that high aptitude participants would benefit more from HV training. Note that previous work had found this interaction both in tests involving generalisation (*Perrachione et al., 2011*) and with trained items (*Sadakata & McQueen, 2014*) so we considered both in our analyses here. There was no evidence of such an interaction in either the Picture Identification or Three Interval Oddity tasks. However, Bayes Factor analyses suggest that the data are inconclusive. We return to these points in the section 'The Role of High Variability Materials in Training and Generalisation' below.

Another finding from the Three Interval oddity test that is worth noting, although it did not concern our hypotheses, is that some trial types were harder than others. Recall that this test involved participants hearing three different stimuli each produced by a different speaker, which makes noting the similarity across two of the stimuli much harder—something we discovered in pilot work, where even before training participants were near ceiling with an equivalent task in which the same speaker produced all three stimuli within a single trial. However, analyses of *trial-type* demonstrated that participants were additionally affected by the gender of the three speakers producing each of the stimuli. Specifically, at pre-test, participants showed best performance for trials where one of the speakers was male and the other two were female, and the target 'odd man' was the male speaker ('easy' trials). In contrast, they showed worst performance if there was one male and two female speakers, but the 'odd man' was one of the female speakers ('hard' trials). Middle level performance was shown for trials where all three speakers were female ('neutral' trials). This is presumably due to participants relying on perceptual cues associated with speaker gender to do the task. Interestingly, our analyses showed that performance only increased for the trials where the odd one was not the lone male (the 'neutral' and 'hard' ones), but not for those where the male was the odd man. Given that participants are not near ceiling at pre-test (67%), it is perhaps surprising that their trained knowledge of the tone contrasts does not boost their performance. One possibility is although they are now better able to use tone cues, they are also *less* likely to use gender based cues, which they may now realise are less reliable, masking improvement based on tone for these particular test items.

## Production tasks

In this study, we used two production tasks, a word repetition task administered pre and post training, in which participants repeated back Mandarin words, and a Picture Naming task testing vocabulary recall, which was administered at post-test only. HV perceptual training for tones has been previously found to transfer to production (*Bradlow & Pisoni, 1999*; *Zeromskaite, 2014*), however the benefits of HV and LV training have not been contrasted.

In the Word Repetition task, there was a significant, though relatively modest improvement in participants' ability to reproduce the tone of the stimuli, such that it could be identified by a native speaker (from pre- to post-test: 70–76%) and in the Picture Naming task, participants showed an ability to recall and reproduce the correct tone, although unsurprisingly with less accuracy than in the repetition task (50%). For Word Repetition, we were also able to look at transfer to untrained words: As in the perception tasks, there was once again equivalent improvement for both trained and untrained items. Together, these results provide evidence that purely perceptual training on tone contrast can transfer to production, as well as to novel items.

In addition to looking at the production of *tones*, we also looked at participants' ability to produce the correct segmental phonology (pinyin-score). Participants showed a small but significant improvement on this measure in Word Repetition (54% correct at pre-test, 58% at post-test), and some ability to recall the segments in the Picture Naming test (50% correct). This indicates some learning of segmental phonology due to training, despite the fact that the focus of the training task was on training tonal information through the presentation of tonal minimal-pairs.

Turning to the role of variability, the predicted benefit of HV training was *not* evident in any of the measures in either of the production tasks, with Bayes factor analyses indicating substantial evidence for the null except for the Word Repetition pinyin-measure, where the evidence was ambiguous. With regard to aptitude, although performance on the Pitch Contour Perception Test at pre-test was predictive of participants' ability to produce tones in both tasks (indicating a relationship between participants perceptual and production ability), we did *not* find the predicted interaction between aptitude and variability condition in either task. Here however, Bayes Factor analyses suggests that the results are inconclusive. We return to these points about variability below.

## The role of high variability materials in training and generalisation

In the current study, across all of the different tests, we did not find either an overall benefit of exposure to HV training materials for generalisation, or any interaction between such a benefit and individual aptitude.

We consider first the lack of *overall* variability benefit for generalisation. Importantly, in addition to finding a pattern of null results (i.e. $p < 0.05$) in the frequentist analyses, additional Bayes Factor analyses also found substantial evidence for the null (BF < 0.33) in all but one of the test measures (Word Repetition, Pinyin, where BF = 0.421). Thus, there is good evidence that, at least for these training and test materials, exposure to stimuli from multiple speakers does *not* lead to greater generalisation in either

perception or production. This finding is consistent with the lack of a main effect of variability condition in the transfer tasks in either *Sadakata & McQueen (2014)* or *Perrachione et al. (2011)* (see also footnote 1). However it is at odds with other phonetic training studies focused on segmental contrasts (*Clopper & Pisoni, 2004*; *Logan, Lively & Pisoni, 1991*; *Lively, Logan & Pisoni, 1993*; *Sadakata & McQueen, 2013*) and with the literature demonstrating a HV benefit in vocabulary learning (*Barcroft & Sommers, 2005, 2014*; *Sommers & Barcroft, 2007, 2011*). This suggests that this overall variability benefit may be restricted to segmental rather than tonal phonetic learning, at least for speakers of a non-tonal L1.

It is difficult to reconcile the lack of benefit for vocabulary learning in the picture naming task, given the findings of *Barcroft & Sommers (2005, 2014)* and *Sommers & Barcroft (2007, 2011)*, since this test is quite similar to that used in their experiments. However, one possibility is that this is due to differences in our training set up (i.e. focused on training tonal contrasts) compared with the earlier vocabulary studies. Nonetheless it remains unclear why *tone learning* should be different from other types of phonetic learning in terms of benefiting from talker-variability. Theoretically speaking, in a framework where all cues compete, variation in idiosyncratic speaker-specific cues would be expected to provide key evidence as to which cues are irrelevant to the phonetic contrast in question (*Apfelbaum & McMurray, 2011*; *Ramscar & Baayen, 2013*; *Ramscar et al., 2010*). This raises the question of how participants in our LV condition are able to generalise at all—that is, how can they identify the phonetically relevant cues compared with the idiosyncratic cues associated with the single speaker to which they were exposed? One possibility is that other variation in our materials aided generalisation, in particular in our real word stimuli, each tone-contrast is encountered with multiple consonants and vowels. If item variability also aids generalisation to new speakers, this might explain why we found equivalent generalisation across conditions instead of seeing greater generalisation in the HV conditions (i.e. even the LV condition is really a HV condition, because of the item variability). On the other hand, *Sadakata & McQueen (2014)* also saw generalisation even for their LV condition, and in their study this condition lacked variation in terms of both speakers and phonetic contexts. This suggests that the relevant cues for the tone contrasts may be sufficiently acoustically salient for learners to identify them, even when exposure occurs in limited contexts.

Another possibility—and the one suggested by the findings of *Sadakata & McQueen (2014)* and *Perrachione et al. (2011)*—is that benefits of HV for generalisation are masked by individual differences. In their studies, only high aptitude participants showed a HV benefit, while low aptitude participants did not. It is possible that for lower aptitude participants, the benefits of exposure to varying, idiosyncratic cues are offset by the greater difficulty that these participants have in attuning to the different speakers during training, as discussed above (section 'Tests of Individual Aptitude'). This explanation is supported by evidence from a study by *Goldinger, Pisoni & Logan (1991)* who explored the effect of increasing the processing cost of multi-speaker input in the context of word recall (in the L1). Specifically, they exposed participants to single vs multi-speaker word

lists, manipulating presentations rates. They found that single-speaker lists produced better word recall than multiple-speaker lists at short inter-word intervals (less than 2,000 ms) whereas this effect was reversed for longer inter-word intervals. This suggests that increasing encoding difficulty can remove the benefits of multi-speaker exposure. Relatedly, *Sinkeviciute et al. (2019)* found that young learners have greater difficulty processing multi-speaker training materials in L2 vocabulary learning, and subsequently fail to show a speaker-variability benefit at test. One interpretation of these findings is that age-related capacity limitations may constrain the ability to benefit from speaker variability, supporting the notion that differences in capacity limitations can affect an individual's ability to benefit from multi-talker training.

Returning to the current study, we did not find an interaction between variability-training and learner aptitude. However, it is important to acknowledge the results of our Bayes factor analyses, which did not find substantial evidence in support of the null over H1 (or H1 over H0) for any of the test tasks. This means that we cannot draw conclusions about this hypothesis from the current data. In theory, we could continue collecting data until we had substantial evidence for either H0 or H1. To explore the feasibility of this, we conducted supplementary analyses to estimate the sample size that might be needed to see substantial evidence for the null (based on the assumption that the error term would reduce in proportion to $\sqrt{SE}$). Taking the Picture Identification test (the test most similar to previous studies) our results suggests that it would require $N > 300$—that is, over five times our current sample size. This suggests that this experimental paradigm is not sufficiently sensitive to address this hypothesis.

Given the ambiguity of our findings with regard to the interaction, it is not appropriate to extensively interpret why we do not find the interaction while the previous studies did. However, we note that there are a variety of differences across the studies which could underpin the different findings, if it holds true. For example, the test of individual differences which we use is harder than that used by *Sadakata & McQueen (2014)* since it uses all six Mandarin vowels (whereas the original study used five, without /y/) and all of the Mandarin tones (where Perrachione et al. used three, without Tone 3). This change also means that that we cannot easily contrast the range of participant scores in the two studies and it may be that the spread of ability of our participant is different from theirs. In addition, our training task is potentially harder than both of the previous studies, that is, involving all four tones in the context of natural Mandarin stimuli in the context of a word learning tasks. Finally, we also note that our statistical analyses are different from both of the previous studies in that they took their continuous aptitude measures and turned these into binary factors using a 'cut off', whereas our statistical approach allows us to use them as continuous variables. However, this should in principle make our approach more powerful than in previous studies.

## Future directions

If the interaction between aptitude and training condition reported in *Sadakata & McQueen (2014)* and *Perrachione et al. (2011)* is to have implications for educational materials,

it is important to establish whether it extends to other more naturalistic materials. Given the relatively small samples in these original studies, and the increasing recognition that psychology experiments have been routinely underpowered (*Maxwell, Lau & Howard, 2015*; and see *Vasishth et al. (2018)* for a recent demonstration in the area of reading) and that can lead to increases in both type 1 and type 2 error, we suggest that it would be useful to implement a direct, high powered replication of these previous studies. We note that having a sufficient sample to provide substantial evidence for H1/H0 using Bayesian methods, or to obtain 90% power for frequentist methods, would likely require a much larger sample than is standard in these types of studies. Given the time-consuming nature of these multiple session training studies, moving to online testing may be necessary to make this feasible (see *Xie et al., 2018* for an example of an acoustic training study done over the web), or alternately multi-lab collaboration may be necessary. Note that this would also allow us to see whether the fact that *Perrachione et al., (2011)* found their interaction with *untrained* voices, whereas *Sadakata & McQueen (2014)* found it only for *trained* voices, is a true difference (due to the different paradigms) or due to power. Critically, successful replication would allow us to then extend the paradigms in such a way as to explore the factors above. For example, would increasing the number of tones to use all four Mandarin tones and/or using natural Mandarin stimuli affect the interaction between variability in the input and learner aptitude?

Although direct replication will play a useful role in establishing these effects, we believe that ultimately it will also be important to develop a more nuanced approach to measuring the factors leading to different levels of aptitude both in tone learning, and in other types of phonetic learning. We note that here in addition to not seeing the predicted interaction with variability, we also didn't see interactions between aptitude and training session in any of our tasks, suggesting that our aptitude measure predicted baseline performance on the task and *not* the ability to improve due to training. In addition, the tasks used to measure 'aptitude' are quite similar in nature to the training and test tasks, decreasing their explanatory value. Our ongoing work explores the combined predictive value of a range of measures including measures of attention, working memory and musical ability. Identifying factors which are predictive of aptitude for tone learning has clear implications for teaching and the personalisation of teaching methods.

## CONCLUSION

We trained naive participants on all four Mandarin tones, using real language stimuli embedded in a word learning task. We found improvements in both production and perception of tones which transferred to novel voices and items. We found that learning was greatest for training with a single voice but that training with a single voice vs four voices (whether intermixed or blocked) lead to equal amounts of generalisation. Although learner aptitude predicted performance in most tasks, there was no evidence that different levels of aptitude lead to better or worse learning from different types of training input.

### Funding

This work was supported by the British Academy Small Grant, a grant awarded to Elizabeth Wonnacott and Meghan Clayards (SG111965) and a Social Sciences and Humanities Research Council of Canada grant (SSHRC #435-2016-0747) to Meghan Clayards and Elizabeth Wonnacott. The funders had no role in study design, data collection and analysis, decision to publish, or preparation of the manuscript.

### Grant Disclosures

The following grant information was disclosed by the authors:
British Academy Small Grant: SG111965.
Social Sciences and Humanities Research Council of Canada grant: SSHRC #435-2016-0747.

### Competing Interests

The authors declare that they have no competing interests.

### Author Contributions

- Hanyu Dong conceived and designed the experiments, performed the experiments, analyzed the data, contributed reagents/materials/analysis tools, prepared figures and/or tables, authored or reviewed drafts of the paper, approved the final draft.
- Meghan Clayards analyzed the data, contributed reagents/materials/analysis tools, prepared figures and/or tables, authored or reviewed drafts of the paper, approved the final draft.
- Helen Brown authored or reviewed drafts of the paper, approved the final draft.
- Elizabeth Wonnacott conceived and designed the experiments, analyzed the data, contributed reagents/materials/analysis tools, authored or reviewed drafts of the paper, approved the final draft.

### Human Ethics

The following information was supplied relating to ethical approvals (i.e. approving body and any reference numbers):

UCL Research Ethics Committee approved the study (Project ID Number: 6176/002).

### Data Availability

The raw data and analytic code are available in the Supplemental Files.

### Supplemental Information

Supplemental information for this article can be found online at http://dx.doi.org/10.7717/peerj.7191#supplemental-information.

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
