# Peer review of "The effects of high versus low talker variability and individual aptitude on phonetic training of Mandarin lexical tones"

_PeerJ, doi:10.7717/peerj.7191_

## Round 0.1 · original submission · Major Revisions

Dear authors

Thank you for your submission to PeerJ and apologies for the delay in getting a decision. This was mainly due to unavailability of suitable reviewers. I have now received 3 reviews and all reviewers suggest a major revision of your manuscript. They agree that this work is relevant, interesting and timely, but a number of issues, mainly to do with clarifying the design and method, the theoretical motivation and interpretation of the data, need to be addressed before your manuscript is accepted for publication.

I look forward to receiving a revised submission.

Vesna Stojanovik
Academic Editor

Reviewer 1 ·

Basic reporting

This manuscript investigates the question of how high-variability training and individual aptitude interact in learning of Mandarin tonal contrasts using real Mandarin words. Training manipulations included low variability (one talker), high variability (four talkers, interleaved presentation), and high variability blocked (four talkers, blocked presentation). Aptitude was assessed with a pitch contour perception task and a categorization task of a synthesized tonal continuum. Tone learning was assessed with discrimination, word identification, and production (word repetition and picture naming) tasks. Although all groups improved on the training and testing tasks from pre-post, no advantage was found for the high-variability groups for trained or untrained stimuli, and no interaction was found with aptitude.

The high quality of statistical methods and data analysis was the main strength of the paper: I thank the authors for their careful description of the methods, providing raw data and R scripts, and for using rigorous statistical methods (e.g., mixed effects logistic regression) to analyze their data. However, several major issues with the framing of the study attenuated my enthusiasm for the manuscript, and these are detailed below.

A primary concern of this study is that it lacks a strong theoretical motivation, and I was unconvinced that it was necessary to replicate the Perrachione et al. (2011) findings using real Mandarin words. There may be good reasons to investigate whether high-variability training is beneficial for learning in a more ecologically valid situation, but this needs to be discussed and motivated more strongly. The manuscript could be strengthened by discussing a theory or model that would predict different outcomes of the high-variability training paradigm when using real Mandarin words as opposed to pseudowords. Perhaps there is reason to believe that learning Mandarin segmental phonology and tonal contrasts simultaneously would yield different results than previous studies (e.g., Perrachione et al., 2011). Otherwise, it does not seem that this study adds to the field’s understanding of high-variability training in phonetic learning.

The literature review was thorough; however, the introduction could be substantially shortened and streamlined. Instead of giving such detailed descriptions of previous studies, I would recommend discussing the previous literature in the context of a theoretical model of (phonetic) learning and making the core questions of the study more prominent.

Line 92: Please cite Bradlow et al. (1999) for long-term retention of production improvements.

Experimental design

As discussed above, it is not clear that there is a gap in the literature that necessitates this study, and reframing the introduction could help clarify the motivation for using real Mandarin words in training and why this might impact our understanding of high-variability training or phonetic learning more generally.

Statistical analyses are appropriate for the data collected, and the methods and analyses are explained in enough detail to enable replication. Plots are easy to read and effectively display the data.

In Figure 11 (training data), in the high aptitude group, the HVB and the LV groups look almost indistinguishable from each other, especially for the first few days. However, in the low aptitude group, the LV group looks like they are performing better than the HVB group. Although I agree that using a continuous measure for aptitude gives you more power in the analysis, it might be worth looking at the aptitude variable categorically after all, especially to compare the HVB-trained participants in high- and low-aptitude groups in the training task.

Validity of the findings

The discussion section should address in greater detail why the current study did not replicate previous high-variability training studies. I agree that some of the earlier studies were likely underpowered; however, this effect has been replicated several times since then with larger sample sizes (e.g., Lively et al., 1994; Perrachione et al., 2011) and for different speech tasks such as accent adaptation (Bradlow & Bent, 2008) and tone learning (Perrachione et al., 2011). The greatest benefits observed in previous studies of high-variability training have typically been for generalization. Perhaps there was no benefit for generalization in the current study due to the pre-training differences in the trained and untrained talkers. Please address.

A discussion on why variability may not be universally beneficial could improve the discussion section, as well, seeing as the LV condition outperformed the HV and HVB conditions on some tasks.

Although some of the original predictions were not borne out, I think there are still some valuable findings in this dataset. For example, the fact that participants improved from pre- to post-test on the pitch contour perception task may have methodological relevance, as this task is often used as an aptitude measure (i.e., Is it really measuring aptitude or is it measuring some type of experience?). A more general discussion of this transfer of learning (from training task to PCPT) would also strengthen the discussion.

The discussion could be further improved by suggesting specific studies that would need to be done to disentangle some of the inconclusive results of the study.

Additional comments

In general, this manuscript needs to be thoroughly proofread. I have noted several typos below, but this list is not exhaustive. There were also inconsistencies in British and American English spellings (e.g., generalization vs. generalization).

Typo line 38
Line 161: productions with an s
Line 225: should be en-dash, not hyphen
Line 275: “Those used in…”
Typo line 285: tones
Typo line 293: that the last two tasks?
Typo line 294: into
Line 378: on-screen, not on- screen
Line 433: main stimulus set?
Line 454: pictures
Typo line 464: space
Typo line 465: conditions
Typo line 469: conditions, but should this actually say, “for this condition”? It seems the low variability group wouldn’t need speakers to be randomized because they only have one.
Line 503: colon not appropriate there. Only use after a complete sentence.
Line 507: no comma after analyses
Line 523: ensured?
Line 551: (2013) in parentheses
Line 806: typo—measures
Line 812: English speakers’
Line 963: en dash, not hyphen
Descriptions for Figures 11 and 12 seem to have gotten cut off (at least in my copy of the manuscript).

Reviewer 2 ·

Basic reporting

Overall, the manuscript was fairly clear. There were some ambiguities (some of which were listed below) to be clarified.
Figures and Tables were appropriate and useful. For ease of processing, it would be great if the tests were itemized in the same order in Session 1 and 8, and that the authors could indicate that Picture Identification and Picture Naming were additional tests.
Table 2 was useful to provide the readers with an overview of the talker manipulation. Given the number of tests administered in the current study, it was difficult for the readers to keep track the purpose of various tests and its relevance to the research questions and predictions, especially when some of them were administered both pre and post-tests; whereas others were only administered post-test. Perhaps the authors would consider including such information in Table 2.
The current ms organized the results (also the procedures) around each administered test. This made it difficult to relate the results to the research questions and predictions. It would be great if the authors could re-organize the stimuli, procedures and results according to the purpose of these tests. For instance, measures of aptitude assessment (PCPT, CSTC); measures of learning (word repetition, 3-level-oddity, picture naming, picture identification). Within the measure of learning, the 4 tests can be further sud-divided.

Experimental design

Line 249-263: the current study tested the benefit of learning from multiple voices, using new paradigm and real Chinese words. It was a partial replication of previous studies on the beneficial effect of learning from multiple voices, as well as the interaction between individual aptitude and learning from multiple voices. Apart from replication, the question was not very well-defined.
The ms discussed the 2 studies (Perrachione et al., 2011; Sadakata & McQueen, 2014) and used their findings to motivate the research questions in the current study. Yet the current study did not point out that Perrachione et al. only manipulated familiar vs novel speakers. That is, the same 18 test items were used in pre-test, training and post-test. Training effect (single talker vs multiple talkers) was generalized to test items from novel speakers. On the other hand, Sakadata & McQueen (2014) manipulated the critical (test) tones along a few dimensions so that the bi-syllabic tone sequences in training was different from those at test. These dimensions included a new position of the critical tone in the bi-syllabic sequence, a new contextual tone in combination with a critical tone to form a bi-syllabic tone sequence, a new vowel bearing the critical tone, the critical bi-syllabic tone sequence from a novel speaker and the critical tone sequence in a sentence context. These differences raised questions as to what kind of information is being learned during variability training and how these information can be generalized across different types (i.e. speaker-variability, item-variability). It would have been useful if the experimental design was clearly tabulated: what was varied, what was kept constant during pre-training, training and post-training, for example:
Aptitude Assessment
PCPT (pre & post-test)
CSTC (pre & post-test)
Assessment
(in the following 2 tests there was a factor of ‘item-novelty’)
Word Repetition (pre & post-test)
Three-interval-oddity (pre & post-test)
Picture identification (post-test) – in familiar voice vs. novel voice
Picture naming (post-test)

When I grouped the stimuli in Appendix A as below,
Training Items
Onset consonants
T1-T2: Ch, m, q, T1-T3: j, sh, x, T1-T4: b, h, zh, T2-T3: b, w, n, T2-T4: d, x, y, T3-T4: d, m, y
Vowels
T1-T2: Ua, ao, ia, T1-T3: iao, u, ue, T1-T4: a, ua, u, T2-T3: i, a, iu, T2-T4: i, ie, u, T3-T4: a, i, a

Non-training items
Onset consonants
T1-T2: sh, t, t, T1-T3: m, g, zh, T1-T4: d, k, x, T2-T3: ch, h, y, T2-T4: m, sh, w, T3-T4: b, d, j
Vowels
T1-T2: i, uo, i, T1-T3: a, ui, e, T1-T4: en, u, ian, T2-T3: i, u, u, T2-T4: o, e, a, T3-T4: ao, ao, ian
It is clear that there was segmental variability for each tone pair contrast and that there was segmental variability between training and non-training items. So, what was it that one expected learners to generalize to? And what was it that high variability training would benefit learners in this regard?

Validity of the findings

The overall result of the current study was that high variability training did not benefit learning Chinese tones and the variability training did not interact with aptitude assessment. Given my concerns about what kind of generalization and benefit that high variability training would bring to various tasks/tests, I was not convinced about the interpretation of the findings in view of the previous literature (e.g. Lively et al., 1993; Logan et al., 1991 etc).

Additional comments

Line 38. Please change ‘This task is particularly difficulty…’
Line 59. Please change ‘… a phonemic contrasts…’
Line 117. ‘… however for the identification task the improvement was greater following high variability training’ was a dangling phrase. Please change.
Line 152 – 153, … 60 NEW words produced by … training speakers, and another with additional 60 new words by new speaker’ Confusing with the word ‘new’. Confusing. There were 2 generalization tasks: one to novel stimuli from familiar speaker, the other to novel stimuli from novel speaker.
Line 143. ‘they do not use pitch information lexically…’ please note that pitch can be used to mark stress at the lexical level too (e.g. ‘import vs. im’port).
Line 163-164. Please note that this was not what Wang et al. 2003 reported (see p. 1037 to 1039: All four tones improved in production post-test, and p.1040 the para starting ‘in terms of individual tones…’ it only referred to T1).
Line 191. Please specify in what way the participants with different aptitude benefited from the different training conditions. High aptitude participants benefited in terms of generalization. Low aptitude ones benefited in terms of performance during training.
Line 201, shouldn’t the correct reponse be ‘1’ instead of ‘2’?
Line 205. Please make it clear that Sadakata et al did the following: Variability was increased in terms of the number of different speakers and the number of different vowels used in bisyllabic sequence.
Line 221-3. This was confusing. Please specify the tasks the benefits were found.
Line 234. Please specify the task the benefits were observed.
Line 236 – 240. Please clarify about the blocked presentation in Sadakata et al. Speaker variability was blocked by session, but not segmental variability. In Perrachione et al, the phonetic variability (i.e. segments) were kept constant and talker variability was varied. Note that Sakadata et al. additionally manipulated item-variability; whereas Perrachione et al. did not. So it is unclear whether trial-by-trial inconsistency is less in the former than the latter.
Line 252. Please note that Wang et al (1999) also used real words in their investigation of variability on learning L2 contrasts.
Line 294. ‘… tapping into mechanisms…’
Line 351. ‘… talker set 2..’
Line 333-338. This was very difficult to process.
Line 412. The choice of ‘different’ stimulus was ambiguous. The test item was different in terms of speakers too. I presume what the authors meant was ‘the odd one out’.
Line 454. ‘… the two pictures always…’
Line 464. ‘… resulting in 288..’
Line 507. ‘ These analyses explore… ‘
Line 523. ‘… which ensured…’
Lines from 526. The description of easy, neutral and hard trial types was not easy to follow. Did the difficulty lie in different voices/talker or different critical/test words?
Line 539. ‘ and in order..’ grammatical issue.
Section 3.2.1 please stick to the PCPT –CSTC order as before.
Line 600. Please change to ‘… in pinyin..’
Line 606. 3 measures were taken. But it stated that the inter-rater reliability was examined for ‘both’ measures. Which ones? Why 2, not 3? This was ambiguous, especially when later on the authors did talk about 3 measures. Please clarify.
Line 630-632. How good were they at pre-test?
Line 649-652. Please clarify ‘ … variability condition (test session by LV…’ The statistical result did not correspond to the statement.
Line 754. ‘participants in the LV groups performed showed no significant difference..’ Please rephrase.
Line 757, Please state that there was no significant difference in pinyin accuracy between variability conditions.
Line 765-766. Why would one expect improved performance, if the task required them to recognize a picture, when the focus of training was on speaker variability? The segmental and f0 information would allow the L2 learners to do the job. From the figures, it looks like the accuracy rate was fairly high overall.
Line 838. ‘… input is in line with…’
Line 851-854. Statements did not correspond to the reported results.
Line 868.’ Since all of the speakers in these test items were new, improvement in this test indicates generalization over speakers.’ I did not follow. The 3-interval-oddity task was administered pre- and post- training.
Line 875. ‘… best performance…’
Line 882. ‘… but not for those…’
Line 898-900. But line 647 reported no improvement after training in the word repetition task. Please clarify.
Line 904. ‘… in the different types of variability conditions..’
Line 909. ‘ 47% ; ‘
Line 919. ‘… is in line with the lack of a main effect in previous tone-training studies..’ Which one? Please specify.
Line 931. ‘… there is a need…’

Reviewer 3 ·

Basic reporting

The English language is generally clear, although there are many spelling errors that should have been corrected prior to submission. I have listed many of them in my line-by-line comments, but the authors need to put more care in proofreading the document.

In general, the paper could do with some streamlining for clarity. It comes across like a dissertation, and this impression is further enhanced when the reader encounters figure labelling in text that seems to match a dissertation rather than this journal manuscript. The authors could cut the manuscript down by a couple of thousand words by looking carefully at the flow of information and critically assessing the relevance of all points. For example, Sadakata and McQueen is reviewed first at line 109, then again on line 192 (as if it is being introduced for the first time).

Do the authors have permission to include the pictures in Appendix A? On line 346 the authors state that they are from "free online clipart databases", but do the licenses of these sites allow for republication?

Raw data are supplied, but the RMarkdown cannot be executed because the following files have not been supplied: CSTC-curve.csv, Word Repetition.csv, Picture Naming.csv.

The data presented in Discrimination.csv are a little confusing because they do not appear to preserve the original order of the stimuli. They are listed as Same1, Same2, and Diff, but there is no record of which position each stimulus was presented in.

Experimental design

The research is relevant, although the authors have not convincingly explained why they made the methodological choices they did. That is, they described the differences between their study and those of Perrachione et al. and Sadakata and McQueen, but they did not justify those choices (line 249). Why is it beneficial to use real Mandarin words instead of pseudowords? What differences do the authors predict in their results due to their inclusion of the fourth tone? Why did the authors choose a word learning task, and how might this affect the predictions, relative to previous studies?

The investigation appears to have been performed to a high technical standard. The methods are certainly described in detail. This could be streamlined somewhat.

Validity of the findings

In a number of the figures, the violin plots suggest that there is at least one participant scoring zero on one or more of the conditions (e.g., Figure 4, Figure 5). Looking at the raw data (in Production_all.csv), there appears to something odd going on with Subject 48. Much of the data for Subject 48 has been coded as 0. Are these missing data? It seems that these 0 values have been included in the analysis, which may explain the apparent outliers in Figure 4. There appear to be other cases in that data file where there are zero values (e.g., over 360 lines of data have "Tone" coded as 0). If these zero values have been included in the analysis, then all of the results could be incorrect. The authors must clarify this as it suggests that the data presented may not be robust.

Additional comments

I commend the authors for undertaking this interesting study. There is a lot to read, which makes the text a little difficult to digest, but the research is timely. The failure to observe an effect of variability is problematic. To suggest that previous findings may be incorrect, on the basis of a null result, is over-interpreting the data. There is a lot of work involved in providing all of that raw data, and I appreciate the RMarkdown. It would be helpful if the data were provided in a format that makes it easy for the reader to recreate the analyses. As it turns out, the results include some aspects that need further clarification, in particular the apparently floor performance of at least one participant in a couple of tasks. My own inspection of the raw data raises the question of whether this apparently floor performance may be due to the inclusion of missing data in the analysis. I was unable to check this in R because the RMarkdown required data files that were not provided. I hope that the authors are able to clarify that aspect. If the data presented are a true reflection of performance, then the authors will need to explain the apparently poor performance of that participant in Figures 3 and 4.

While reading the document I made line-by-line comments. I hope that these assist the authors in further revisions of the manuscript.

Line-by-line comments:
Abstract, lines 29-30. "different variability training" is vague. Does this mean that the two types of high-variability training do not differ from each other, or that there was no difference between high and low?

line 38. Grammar - "this task is particular difficulty"
line 39. "the same acoustic properties" is vague. Exactly what do you mean?
line 46. Please specify what "it" is.
line 48. "Pisonni"->"Pisoni"
line 70, and throughout. Phonetic symbols do not need to be in italics. In fact, it's a little distracting.
line 74. "to both" -> "both to"
line 82. It is not "so-called" - it is simply the HVPT.
line 104. It sounds like the empirical review that follows is providing examples of studies which have not explicitly tested high vs low variability training, but it seems to be providing examples of studies which have compared the two. This sentence needs to be clarified.
line 118. This paragraph would benefit from a conclusions. It is not clear what the reader is supposed to infer from this review.
line 125. "which" -> "that"
line 127. "(2014) which" -> "(2014), which"
line 131. They show greater learning that what? Please provide the comparison.
line 135. "exists" is clumsy. I suggest rewording this sentence starting with "Lexical tone is…". This will also have the positive effect of putting the topic of the sentence at the beginning.
line 142. "while languages such as English" -> "while non-tonal languages such as English"
line 182. What are the three standard diacritics?
line 200-201. Should the correct response be 1 for this example?
line 211. "continua" -> "continuum"
line 220-221. Choose a consistent spelling for benefited/benefitted.
line 236. Sadaka -> Sadakata
line 238. "in this study" - Do you mean "in that study"?
line 313. add a unit - i.e., 12.6 years.
line 351. taker -> talker
line 359. pitch -> fundamental frequency
line 381. I do not find these abbreviations to be self-evident. They make the text difficult to follow. PCPT is defined many pages earlier and the reader will likely need to turn back to be reminded what it stands for. The CSTC is not defined in the text (only in Table 2 and Figure 1). I suggest spelling the tasks out in full in the text. If not, both abbreviations should be defined in the text.
line 439. White noise would certainly prevent participants from being able to access the item in echoic storage, but I'm not sure that it would prevent rehearsal using the phonological loop. Articulatory suppression would be needed to achieve that.
line 465. condition->conditions
line 513. I thought that LME stood for linear mixed effect model, not logistic mixed effect model.
line 523. ensued -> ensured
line 624. Figure 3.3.1.2. This figure label does not appear to be related to this particular document.
line 658. Figure 3.3.2. Ditto. Many other figure numbers to not match those in this manuscript.
line 663. "accidental" -> "incidental"
line 822. "naming task" -> "picture naming task". A naming task involves reading out letter strings.
line 855, Section 4.2. The authors should refer to Strange and Shafer (2008), who review the varying demands of different discrimination tests (AX, AXB, and Oddity). Strange, W., & Shafer, V. L. (2008). Speech perception in second language learners: The re-education of selective perception. In J. G. Hansen Edwards & M. L. Zampini (Eds.), Phonology and second language acquisition (pp. 159-198). Philadelphia, PA: John Benjamins.
line 863. Something wrong in this sentence around "speakerin"

Table 1. Staring Age -> Starting Age

Figures. Trained vs. untrained needs to be explained in the figure captions where relevant. It gives the reader the impression that there were trained and untrained groups, rather than items that were included in the training set or novel.

Figure 4. There seem to be some outliers in the post-test, with at least one participant apparently scoring zero (if I understand the violin plots correctly). Same in Figure 5.

Figure 6. The order for trained and untrained is reversed here relative to the previous figures. Be consistent.

Figure 10. Why are these violin plots lower quality than those in previous figures?

---

## Round 0.2 · Minor Revisions

The reviewers have pointed out to some minor issues to do with 1) typos (both reviewers have provided a list of these) and 2) some points of clarification on the experimental design by Reviewer 1.

Reviewer 1 ·

Basic reporting

I thank the authors for their careful attention to the reviewer comments and substantial revisions to the paper. I found it to be a much clearer and more enjoyable read this time around.

I also appreciate the authors' commitment to replicability and open science by sharing their data and analysis scripts.

Here are some typos that need to be corrected:

Line 212, typo: generalisationParticipants

Line 234: in particular or particularly

Line 239: what does “s” mean?

Line 288: impact the interaction, rather than impact on

Lines 365-366: ) after Committee) and / after approval/ don’t seem necessary

Line 377: extraneous ) at end of sentence

In line 448 and many others, this text appears: Error! Reference source not found, please check and correct these

Line 842: hyphen should be an en dash

Line 941: should be Perrachione et al. (2011), not 2012; please also make this change in footnote 1

Line 1152: extra set of parentheses

Line 1191: “impact upon” sounds funny--"affect" maybe?

Line 1224: parentheses are closed but not opened

Line 1233-1234: wording is confusing—you say both studies found the interaction with untrained voices but it seems like those should be contrasting statements

Parts of the boxes in Figure 1 seem to be cut off (at least in the pdf copy).

Experimental design

Although some justification for their extension of previous work was provided (mainly for purposes of ecological validity), I would still like for the authors to spell out why they might see differences between non-words and real words more clearly in the manuscript. Ecological validity is a fine motivation, but it seems like there is also motivation from previous literature, which will make this a more scientifically interesting contribution in addition to an educationally relevant contribution. In general I thought the motivation for the current study was better articulated in the rebuttal letter than in the manuscript, and I think the manuscript can be improved by including the following points that were included in the rebuttal letter:
-A more explicit description in the manuscript about the relevant acoustic differences that have been found in non-words vs. real words and why we might expect different results from real word stimuli from the previous studies that have used non-words: For example, a quick skim of the two references cited in the rebuttal letter seem to imply that non-words are sometimes hyperarticulated/some acoustic differences are exaggerated, which could mean non-words are easier for non-native listeners to learn. I did like the justification in lines 277-278 that using real words presents the learner with both non-native segmental and suprasegmental information simultaneously.
-A brief discussion similar to the paragraph from the rebuttal letter starting with "We did not have precise theoretical predictions about how these changes – which generally increase the difficulty of the learning task – might affect the key finding of the previous work..." would improve the current study section of the introduction because it would give the reader a clear idea of some of the potential differences that could be seen from the extensions from previous work that this study carries out.

Validity of the findings

No comment.

Reviewer 2 ·

Basic reporting

no comment

Experimental design

no comment.

Validity of the findings

no comment.

Annotated reviews are not available for download in order to protect the identity of reviewers who chose to remain anonymous.

Reviewer 3 ·

Basic reporting

The basic reporting fine.

Experimental design

The experimental design is fine.

Validity of the findings

The findings appear to be valid. I am not an expert in the Bayesian approach to null hypothesis testing. I am aware that it is becoming more popular, and its inclusion in this manuscript is welcome, but I am not well placed to judge whether it was conducted correctly.

Additional comments

The manuscript reads much more clearly now. The authors have clearly gone to a lot of effort to streamline the presentation and to respond to the reviewers' comments. It certainly does not read like a dissertation anymore. That said, there is still a lot to take in with this paper and I think readers are going to find it difficult to digest. That is a consequence of the complicated design and do not think that there is much more that the authors could do to improve that.

I think the manuscript is nearly ready for publication, but I need to point out that there is still evidence of inattention to detail in this manuscript. I have listed some of that evidence in my line-by-line comments below. I have not checked the data in the supplemental materials for this version. I went to the supplementary materials when reviewing the last version because the figures led me to question the data. I have no reason to inspect the data files this time. Reviewers should not be expected to check whether analyses have been conducted without errors, so I hope that the continued infelicities found in the manuscript are not also evident in the data.

In sum, I commend the authors for undertaking this interesting study. A lot of weight has been given the Lively/Logan/Pisoni/Bradlow papers and we need studies to question the accepted wisdom about the benefit of HVPT.

Line 130. Adaption -> adaptation
Line 212. Something has gone wrong here. Is there something missing?
Line 234. Particular -> particularly
Line 258. Sadakaka?
Line 267. I can't see where those models are discussed. Mentioned, perhaps. But discussed, no.
Line 292. What is "s"? I presume it's "speaker" but this really needs to be spelled out.
Line 448. In future, please check the final submitted manuscript carefully before submitting.
Line 546, 549, 552. In future, please check the final submitted manuscript carefully before submitting.
Line 553. )) -> )
Line 744. ....
Line 1152. Delete ()
Line 1187. Younger than who?
Line 1215. Where as -> whereas

---

## Round 0.3 · accepted · Accept

Dear authors

Thank you for resubmitting your revised manuscript and for addressing the reviewers' comments. I am pleased to inform you that the manuscript has now been accepted for publication.